# Joint Gradient Balancing for Data Ordering in Finite-Sum Multi-Objective Optimization

**Hansi Yang, James Kwok**
Department of Computer Science and Engineering,
Hong Kong University of Science and Technology, Hong Kong, China
`{hyangbw, jamesk}@cse.ust.hk`

## Abstract

In finite-sum optimization problems, the sample orders for parameter updates can significantly influence the convergence rate of optimization algorithms. While numerous sample ordering techniques have been proposed in the context of single-objective optimization, the problem of sample ordering in finite-sum multi-objective optimization has not been thoroughly explored. To address this gap, we propose a sample ordering method called JoGBa (**Jo**int **G**radient **Ba**lancing), which finds the sample orders for multiple objectives by jointly performing online vector balancing on the gradients of all objectives. Our theoretical analysis demonstrates that this approach outperforms the standard baseline of random ordering and accelerates the convergence rate for the MGDA algorithm. Empirical evaluation across various datasets with different multi-objective optimization algorithms further demonstrates that JoGBa can achieve faster convergence and superior final performance than other data ordering strategies.

## 1 Introduction

Many well-known machine learning problems involve jointly optimizing multiple objectives in model training. Examples include multi-task learning (Sener & Koltun, 2018), meta-learning (Ye et al., 2021), learning with fairness and safety constraints (Zafar et al., 2017) and multi-agent reinforcement learning (Moffaert & Nowé, 2014). Mathematically, these problems share the same formulation of minimizing a vector-valued loss function $\mathcal{L}$ and can be defined as:

$$\min_{\boldsymbol{w} \in \mathbb{R}^d} \mathcal{L}(\boldsymbol{w}) \coloneqq [\mathcal{L}_1(\boldsymbol{w}), \ldots, \mathcal{L}_M(\boldsymbol{w})]. \tag{1}$$

Here, each loss function $\mathcal{L}_m(\boldsymbol{w}), m = 1, \ldots, M$ corresponds to a training objective and can be expressed by $\mathcal{L}_m(\boldsymbol{w}) = \sum_{n=1}^{N} \ell_m(\boldsymbol{w}, \xi_n)$, where each $\xi_n$ denotes a training sample and $\ell_m$ is the per-sample loss. Solving problem (1) is fundamentally different from common single-objective optimization problems as different objectives may have conflicts with each other. A straight-forward baseline is to optimize a weighted average of the multiple objectives, also known as *static or unitary weighting* (Kurin et al., 2022; Xin et al., 2022). Its performance then largely depends on how to choose the weights to balance different objectives, and may involve huge amount of tuning efforts. A popular alternative is thus to *dynamically weight* gradients from different objectives to avoid conflicts between them. Generally, these methods share the same procedure: First, compute all the gradients of each objective, then compute a set of weights for different objectives based on their gradients. The model is updated by the weighted sum of all gradients, while the weights can dynamically change. The pioneering work of this approach is the multi-gradient descent algorithm (MGDA) (Désidéri, 2012) and its stochastic variants (Liu & Vicente, 2021; Fernando et al., 2023; Zhou et al., 2022; Chen et al., 2024). Later works further improve upon MGDA by considering the worst improvement among different objectives (Liu et al., 2021; Ban & Ji, 2024), as well as constructing a bargaining game between different objectives (Navon et al., 2022).

While the aforementioned methods can be used to compute weights dynamically based on the loss gradients, a less-investigated issue for the stochastic optimization of finite-sum multi-objective problem is how we should order the samples for the computation of their stochastic gradients so that problem (1) can be solved efficiently. For single-objective optimization in the finite-sum setting,

many methods have been proposed for ordering the samples in stochastic optimization (Ying et al., 2017; Gürbüzbalaban et al., 2019; Lu et al., 2021; Mohtashami et al., 2022; Lu et al., 2022). With multiple objectives, a simple generalization of these methods is to treat the weighted average of all loss gradients as sample "gradient" for update (Figure 1(a)). However, the gradient weights may change drastically during model update, making existing methods unstable and often do not outperform the simple baseline of random ordering. Another simple extension is to run the single-objective sample ordering algorithm on each objective separately, but this can lead to different orderings for different objectives (Figure 1(b)). Moreover, it overlooks possible conflicts between gradients from different samples, and thus may still yield limited improvement over random ordering.

Motivated by the above limitations, in this paper, we propose a novel sample ordering framework for multi-objective optimization. As illustrated in Figure 1(c), the proposed method jointly provides sample orderings for different objectives by solving an online vector balancing problem with the gradients on each objective. The online vector balancing problem allows us to control the maximum norm of total model update in each epoch, leading to the theoretical guarantee of accelerated convergence. Specifically, our theoretical results demonstrate that the proposed method improves over the random ordering baseline for finite-sum multi-objective optimization, with smaller sample variance and faster convergence. Empirical results on a variety of data sets with multiple learning objectives demonstrate that the proposed method achieves faster convergence and better performance than the other data sampling methods.

Our contributions can be summarized as follows:

- We propose a novel data ordering method that uses gradient balancing across different objectives to accelerate convergence.

- We propose a novel theoretical framework to analyze multi-objective optimization with different data ordering for each objective.

- Empirical results on various multi-task learning data sets demonstrate effectiveness of the proposed method.

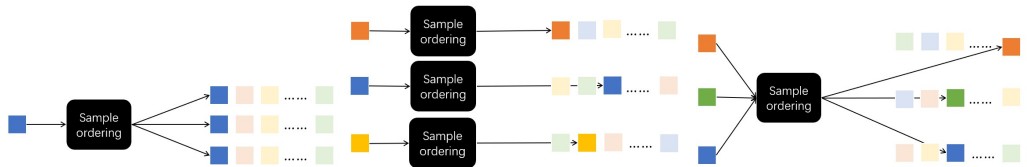

(a) Same ordering for all objectives.    (b) Data ordering on each objective separately.    (c) Proposed method: joint data ordering on all objectives.

Figure 1: Visualization of different data ordering approaches for multi-objective optimization.

## 2 RELATED WORKS

### 2.1 PERMUTATION-BASED SGD FOR FINITE-SUM OPTIMIZATION

Stochastic optimization often assumes that training samples are independently from an underlying distribution in each iteration. However, this assumption does not match with practical implementations that typically use finite training samples in a certain order. Instead, permutation-based SGD proposes to first sort all the training samples by an order, and then use the samples following this order. An example is random reshuffling (Ying et al., 2017) and the related shuffle-once method (Bertsekas, 2011; Gürbüzbalaban et al., 2019), which first generates a random order of all training samples in each epoch, and then uses the training samples following this order in the subsequent iterations. Theoretical analysis of random reshuffling dates back to Recht & Ré (2012). Rajput et al. (2021) introduces a variant of random reshuffling that reverses the order every two epochs, and theoretically demonstrates that this variant achieves faster convergence for quadratic objectives.

Instead of using a random order, Lu et al. (2021),Mohtashami et al. (2022), and Lu et al. (2022) try to find better sample orders. These works are mostly based on the herding problem (Welling, 2009), which minimizes the consecutive errors of stochastic gradients. Theoretical analysis (Cha et al., 2023) demonstrates that the ordering based on the herding problem is asymptotically optimal. There

are different methods to solve the herding problem. Mohtashami et al. (2022) evaluates gradients on all samples first and then solves the herding problem to obtain the order for all samples before starting an epoch. Lu et al. (2021) uses stale gradients from the previous epoch to estimate the gradient on each sample. Lu et al. (2022) proposes to solve the herding problem via online vector balancing, which removes the additional storage cost in (Mohtashami et al., 2022; Lu et al., 2021).

Despite the aforementioned improvements, existing works on permutation-based SGD only focus on single-objective optimization problems. While some simple extensions exist for training with multiple objectives (e.g., by using the weighted gradient or ordering samples for each objective separately), these simple extensions do not always yield much improvements, as will be demonstrated in our empirical results.

## 2.2 GRADIENT-BASED MULTI-OBJECTIVE OPTIMIZATION

To balance the optimization on different objectives, most existing algorithms use the weighted average of all objective gradients to update the model. There are different ways to compute the weights for different objectives. Some works perform the weighting based on some heuristics. Examples include using the prediction uncertainty (Kendall et al., 2017), gradient norms (Chen et al., 2018) or task difficulty (Guo et al., 2018). Another line of works propose to compute the objective weights from some sub-problems on the objective gradients. The pioneering work is MGDA (Désidéri, 2012), which computes the weights by avoiding conflicts across any objective. Stochastic variants of MGDA with optimization convergence guarantees have been proposed in (Liu & Vicente, 2021; Zhou et al., 2022; Fernando et al., 2023; Chen et al., 2024). PCGrad (Yu et al., 2020) proposes to project the gradients of tasks to the normal plane of the other tasks with conflicting gradients. CA-Grad (Liu et al., 2021) searches for an update direction in a neighborhood of the average gradient that maximizes the worst improvement of any task. Nash-MTL (Navon et al., 2022) proposes to look for a fair gradient direction based on a bargaining game between different objectives.

Convergence analysis for the deterministic MGDA algorithm dates back to (Fliege et al., 2019). Later on, stochastic variants of MGDA are introduced (Liu & Vicente, 2021; Zhou et al., 2022; Fernando et al., 2023; Chen et al., 2024). However, the vanilla stochastic MGDA introduces a biased estimate of the dynamic weight, which results in the biased estimate of update direction during optimization. To address this issue, Liu & Vicente (2021) propose to increase the batch size during optimization, and prove the convergence of stochastic MGDA with the Lipschitz continuity assumption on the objective weights $\lambda^*(\boldsymbol{w})$ with respect to the loss gradients $\nabla \mathcal{L}(\boldsymbol{w})$. Nevertheless, as first proved in (Zhou et al., 2022, Proposition 2), this assumption does not hold in general. To address this problem, momentum-based bias reduction algorithms (Zhou et al., 2022; Fernando et al., 2023) are proposed to eliminate such unrealistic assumptions. The convergence of the MGDA algorithm without the unrealistic Lipschitzness assumption is first established in (Chen et al., 2024), which propose to mitigate the bias in update direction via double sampling. Most existing works focus on the convergence analysis under an online setting instead of the finite-sum setting, and ignores the impact of sample orders in their theoretical analysis.

## 3 PROPOSED METHOD

### 3.1 MULTIPLE SAMPLE ORDERINGS FOR MULTIPLE OBJECTIVES

A simple extension of existing single-objective sample ordering methods to multi-objective optimization is to use the weighted average of all loss gradients as sample "gradient", and follow existing data ordering methods on the weighted gradient. When the objective weights do not change with different samples, this extension can be regarded as using the weighted objective as the only objective in the existing methods. However, since the objective weights are constantly changing, using the same sample order cannot well tackle the possible conflicts between different objectives.

To alleviate this problem, we propose to use different sample orders for different objectives. Specifically, for a data set with $K$ samples, we generate an order $\pi_t^m : \{1, \ldots, K\} \to \{1, \ldots, K\}$ for the $m$-th objective. Some simple methods to generate the order $\pi_t^m$ in each epoch $t$ are listed below:

1. **Random**: In each epoch $t$, the data sets are randomly shuffled to generate an ordering $\pi_t^m$ for each objective.

2. FlipFlop: For each objective, create a new order $\pi_{t+1}^m$ by reversing the previous $\pi_t^m$, i.e., $\pi_{t+1}^m(k) = \pi_t^m(K + 1 - k)$.

3. Random FlipFlop, which performs Random on even epochs and FlipFlop on odd epochs.

## 3.2 SAMPLE ORDERING BY ONLINE VECTOR BALANCING

Besides the simple ordering methods in Section 3.1, some recent works (Lu et al., 2021; Mohtashami et al., 2022; Lu et al., 2022) propose to adaptively find a good order for all training samples in each epoch for faster convergence of stochastic optimization with a single objective. An example is GraB (Lu et al., 2022), which tries to find a sample ordering $\pi$ that minimizes the maximum norm of parameter update in each epoch, i.e., $\max_{K'} \|\boldsymbol{w}^{(K')} - \boldsymbol{w}^{(1)}\|_\infty$. With a single objective $\ell$, the model parameters are updated by $\boldsymbol{w}^{(k+1)} = \boldsymbol{w}^{(k)} - \alpha \nabla \ell(\boldsymbol{w}, \xi_{\pi(k)})$ at each iteration $k$ in an epoch. This problem is then transformed to the online vector balancing problem defined below:

**Definition 3.1** (Online Vector Balancing (Spencer, 1977)). Given $K$ vectors $\{\boldsymbol{z}_k\}_{k=1}^K \in \mathbb{R}^d$, arriving one at a time, the goal of *online vector balancing* is to assign a sign $\epsilon_k \in \{-1, +1\}$ to each vector upon receiving it so as to minimize $\max_{m \in \{1,...,K\}} \|\sum_{k=1}^m \epsilon_k \boldsymbol{z}_k\|_\infty$.

We propose to generalize this problem to multiple objectives by replacing the gradients on a single objective to those on multiple objectives and jointly consider their influence to the model updates. The complete procedure, called JoGBa (**Jo**int **G**radient **Ba**lancing), is shown in Algorithm 1. Specifically, at the $k$-th iteration of epoch $t$, we compute the gradients $\{\nabla \ell_m(\boldsymbol{w}_t^{(k)}, \xi_{\pi_t^m(k)})\}_{m=1}^M$ for all $M$ objectives w.r.t. the current model parameter $\boldsymbol{w}_t^{(k)}$. The sample order $\pi^m$ for each objective is then determined by solving the balancing problem on the gradients from different objectives, implemented by routine Balancing in step 11. While there exists different ways to solve the online vector balancing problem and compute the gradient sign $\epsilon_{m,k,t}$, here we follow GraB and use a greedy algorithm that works well in practice. As shown in Algorithm 2, we compare two vector norms $\|\boldsymbol{s} + \boldsymbol{g}_{m,k,t}\|_\infty$ and $\|\boldsymbol{s} - \boldsymbol{g}_{m,k,t}\|_\infty$, where $\boldsymbol{s} + \boldsymbol{g}_{m,k,t}$ corresponds to putting this sample at the beginning and $\boldsymbol{s} - \boldsymbol{g}_{m,k,t}$ corresponds to putting this sample at the end. Since the online vector balancing problem in Definition 3.1 tries to minimize the vector sum's norm, we choose the sample order that can lead to the smallest norm, as is indicated by the value of $\epsilon_{m,k,t}$. The vector $\boldsymbol{s}$ is shared among different objectives to enable joint balancing across their corresponding gradients. After the balancing routine is complete, we compute the objective weights $\lambda$ by any multi-task learning algorithm (routine MTL), such as MGDA (Désidéri, 2012) or Nash-MTL (Navon et al., 2022). We then update the mean $\boldsymbol{v}$ of all gradients and perform model update on $\boldsymbol{w}_t^{(k)}$.

## 3.3 THEORETICAL ANALYSIS

In this section, we theoretically demonstrate how Algorithm 1 improves upon simple extensions of sample ordering methods to multi-objective optimization. Since the convergence analysis of multi-objective optimization is different from optimizing a single objective, we first introduce the definition of Pareto stationary. Denote the gradients for all $M$ objectives as $\nabla \mathcal{L}(\boldsymbol{w}) \in \mathbb{R}^{d \times M}$, where $\mathcal{L}(\boldsymbol{w})$ is defined as in (1), and define $\Delta^M$ as the following set:

$$\Delta^M := \left\{ \lambda \in \mathbb{R}^M : \sum_{m=1}^M \lambda_m = 1, \lambda_m \geq 0, \forall m = 1, \dots, M \right\}.$$

Analogous to the stationary and optimal solutions for a single objective, we define Pareto stationary and Pareto-optimal solutions for the multi-objective optimization problem $\min_{\boldsymbol{w} \in \mathbb{R}^d} \mathcal{L}(\boldsymbol{w})$.

**Definition 3.2** (Pareto stationary and Pareto optimality (Momma et al., 2022)). If there exists a convex combination of the gradient vectors that equals to zero, i.e., there exists $\lambda \in \Delta^M$ such that $\nabla \mathcal{L}(\boldsymbol{w})\lambda = 0$, then $\boldsymbol{w} \in \mathbb{R}^d$ is *Pareto stationary* for $\mathcal{L}$. If there is no $\boldsymbol{w} \in \mathbb{R}^d$ and $\boldsymbol{w} \neq \boldsymbol{w}^*$ such that, for all $\mathcal{L}_m(\boldsymbol{w})$ defined in (1) with $m = 1, \dots, M$, $\mathcal{L}_m(\boldsymbol{w}) \leq \mathcal{L}_m(\boldsymbol{w}^*)$, and for at least a $m' = 1, \dots, M$, $\mathcal{L}_{m'}(\boldsymbol{w}) < \mathcal{L}_{m'}(\boldsymbol{w}^*)$, then $\boldsymbol{w}^*$ is *Pareto-optimal* for $\mathcal{L}$.

By definition, at a Pareto stationary point, there is no common descent direction for all objectives. A necessary and sufficient condition for $\boldsymbol{w}$ being Pareto stationary for smooth objectives is that $\min_{\lambda \in \Delta^M} \|\nabla \mathcal{L}(\boldsymbol{w})\lambda\| = 0$ (Tanabe et al., 2019), which corresponds to the stationary condition

---

**Algorithm 1** JoGBa: **Jo**int **G**radient **Ba**lancing for Multi-Objective Optimization.

---

1: **Input:** number of epochs $T$, initialized order $\pi_1$, initialized weight $\boldsymbol{w}_0$, stale mean $\boldsymbol{v}_0 = \boldsymbol{0}$, step size $\alpha$.
2: **for** $t = 0, \ldots, T-1$ **do** {$t$ is the number of epochs}
3:     **for** $m = 1, \ldots, M$ **do** {$m$ is the index on different objectives}
4:        Initialize left index $l_m \leftarrow 1$, right index $r_m \leftarrow K$
5:     **end for**
6:     Initialize running average $\boldsymbol{s} \leftarrow \boldsymbol{0}$, stale mean $\boldsymbol{v}_{t+1} \leftarrow \boldsymbol{0}$.
7:     **for** $k = 1, \ldots, K$ **do** {$k$ is the number of iterations in each epoch, $\pi_t^1(k), \ldots, \pi_t^M(k)$ indicates the sample index we select for each objective}
8:        Sample data $\xi_{\pi_t^1(k)}, \ldots, \xi_{\pi_t^M(k)}$ from data set $\mathcal{D}$
9:        **for** $m = 1, \ldots, M$ **do** {Compute the gradient on the $m$-th objective and updates its sample order $\pi_{t+1}^m$ for next epoch $t+1$}
10:           Compute gradient $\nabla \ell_m(\boldsymbol{w}_t^{(k)}; \xi_{\pi_t^m(k)})$ and centered gradient $\boldsymbol{g}_{m,k,t} \leftarrow \nabla \ell_m(\boldsymbol{w}_t^{(k)}; \xi_{\pi_t^m(k)}) - \boldsymbol{v}_t$
11:           Compute sign for the current gradient: $\epsilon_{m,k,t} \leftarrow \texttt{Balancing}(\boldsymbol{s}, \boldsymbol{g}_{m,k,t})$
12:           **if** $\epsilon_{m,k,t} = +1$ **then**
13:             Update $\boldsymbol{s}$ and left index $l_m$: $\boldsymbol{s} \leftarrow \boldsymbol{s} + \boldsymbol{g}_{m,k,t}$; $\pi_{t+1}^m(l_m) \leftarrow \pi_t^m(k)$; $l_m \leftarrow l_m + 1$.
14:           **else**
15:             Update $\boldsymbol{s}$ and right index $r_m$: $\boldsymbol{s} \leftarrow \boldsymbol{s} - \boldsymbol{g}_{m,k,t}$; $\pi_{t+1}^m(r_m) \leftarrow \pi_t^m(k)$; $r_m \leftarrow r_m - 1$.
16:           **end if**
17:        **end for**
18:        Compute weights $\lambda$ from multi-task learning algorithms $\lambda = \text{MTL}(\{\nabla \ell_m(\boldsymbol{w}_t^{(k)}; \xi_{\pi_t^m(k)})\}_{m=1}^M)$
19:        Update stale mean $\boldsymbol{v}_{t+1} \leftarrow \boldsymbol{v}_{t+1} + \frac{1}{K} \sum_{m=1}^M \nabla \ell_m(\boldsymbol{w}_t^{(k)}; \xi_{\pi_t^m(k)})$
20:        Optimizer Step: $\boldsymbol{w}_t^{(k+1)} \leftarrow \boldsymbol{w}_t^{(k)} - \alpha \sum_{m=1}^M \lambda_m \nabla \ell_m(\boldsymbol{w}_t^{(k)}; \xi_{\pi_t^m(k)})$
21:     **end for**
22:     Use the model parameter from last iteration as the initialization for next epoch $t+1$: $\boldsymbol{w}_{t+1}^{(1)} \leftarrow \boldsymbol{w}_t^{(K+1)}$.
23: **end for**

---

**Algorithm 2** Online greedy implementation of $\texttt{Balancing}(\boldsymbol{s}, \boldsymbol{g}_{m,k,t})$.

---

1: **Input:** $\boldsymbol{s}, \boldsymbol{g}_{m,k,t}$.
2: $\epsilon_{m,k,t} = 1$ if $\|\boldsymbol{s} + \boldsymbol{g}_{m,k,t}\|_\infty \leq \|\boldsymbol{s} - \boldsymbol{g}_{m,k,t}\|_\infty$ else $\epsilon_{m,k,t} = -1$.
3: Return $\epsilon_{m,k,t}$.

---

$\|\nabla \mathcal{L}_m(\boldsymbol{w})\| = 0$ for a specific objective $\mathcal{L}_m$. Then, similar to the gradient norm $\|\nabla \mathcal{L}_m(\boldsymbol{w})\|$ for single-objective optimization, the quantity $\min_{\lambda \in \Delta^M} \|\nabla \mathcal{L}(\boldsymbol{w})\lambda\|$ can be used as a measure of Pareto stationarity (Désidéri, 2012; Fliege et al., 2019; Liu & Vicente, 2021; Fernando et al., 2023).

Now we list several assumptions that are necessary for our theoretical results. These assumptions are all commonly used in previous theoretical analysis (Liu & Vicente, 2021; Fernando et al., 2023; Zhou et al., 2022; Chen et al., 2024) on the convergence of multi-objective optimization methods:

**Assumption 3.3** (Lipschitzness of $\ell_m(\boldsymbol{w})$'s and $\mathcal{L}(\boldsymbol{w})$)**.** For all $m \in \{1, \ldots M\}$, $\ell_m(\boldsymbol{w}, \xi)$ is $f$-Lipschitz continuous for all training samples $\xi$. $\mathcal{L}(\boldsymbol{w})$ is then $F$-Lipschitz continuous in the Frobenius norm with $F = \sqrt{M}f$.

**Assumption 3.4** (Lipschitz smoothness of $\ell_m(\boldsymbol{w})$'s and $\mathcal{L}(\boldsymbol{w})$)**.** The gradient $\nabla \ell(\boldsymbol{w}, \xi)$ is $f_1$-Lipschitz continuous for all $m \in \{1, \ldots, M\}$ for all $\xi$. $\nabla \mathcal{L}_m(\boldsymbol{w})$ is then $F_1$-Lipschitz continuous in the Frobenius norm with $F_1 = \sqrt{M}f_1$.

**Assumption 3.5** (Bounded gradient variance for each objective)**.** For any $\boldsymbol{w}$ and sample $\xi$, the $m$-th loss function satisfies $\|\nabla \ell_m(\boldsymbol{w}, \xi) - \nabla \mathcal{L}_m(\boldsymbol{w})\|_2^2 \leq \sigma_m^2$ for some given $\sigma_m$.

We then have the following convergence result if we use the MGDA algorithm (Désidéri, 2012; Sener & Koltun, 2018) to compute the objective weights $\lambda$. Proof is in Appendix A.1.

**Theorem 3.6** (Random Ordering)**.** *Suppose Assumptions 3.3, 3.4 and 3.5 hold. Define* $\Delta = \max_{\lambda \in \Delta^M} \mathcal{L}(\boldsymbol{w}_0)\lambda - \min_{\boldsymbol{w} \in \mathbb{R}^d, \lambda \in \Delta^M} \mathcal{L}(\boldsymbol{w})\lambda$ *as the maximum difference between objective values at initialization and that at Pareto optimality. Consider the model parameters[1] $\{\boldsymbol{w}_t^{(1)}\}$ generated by the MGDA algorithm with random sample ordering. Set* $\alpha = \sqrt{\frac{2\Delta}{F_1(F^2 + \sigma^2)KT}}$ *where*

---

[1]Superscript 1 indicates the model parameters at the beginning of each epoch.

$\sigma^2 = \max_m \sigma_m^2$ with $\sigma_m^2$ defined in Assumption 3.5, then,

$$\frac{1}{T} \sum_{t=0}^{T-1} \mathbb{E}\left[\min_{\lambda \in \Delta^M} \|\nabla \mathcal{L}(\boldsymbol{w}_t^{(1)})\lambda\|^2\right] \leq \sqrt{\frac{2F_1\Delta(F^2 + \sigma^2)}{KT}} + \frac{\sigma^2(1 + \log(T))}{T}. \qquad (2)$$

To analyze the convergence rate of Algorithm 1 that uses online gradient balancing to determine sample ordering for different objectives, we first need an additional assumption on the `Balancing` subroutine, which is also used for gradient balancing with single objective in (Lu et al., 2022).

**Assumption 3.7.** (Balancing Bound) For the subroutine `Balancing` in Algorithm 1, denote its input vectors as $\boldsymbol{z}_1, \ldots, \boldsymbol{z}_n \in \mathbb{R}^d$ which satisfy $\|\boldsymbol{z}_i\|_2 \leq 1, \forall i = 1, \ldots n$. Suppose the subroutine assigns each vector $\boldsymbol{z}_i$ a sign $\epsilon_i \in \{-1, +1\}$. There exists a constant $A > 0$ such that $\|\sum_{i=1}^k \epsilon_i \boldsymbol{z}_i\|_\infty \leq A$ for all $k \in \{1, \ldots, n\}$.

From Definition 3.1, solving the online vector balancing problem corresponds to minimizing $A$ in Assumption 3.7. We also have the following Proposition that controls the maximum norm of parameter updates in each epoch. Proof is in Appendix A.2.

**Proposition 3.8.** *Under Assumption 3.3 and 3.7 Algorithm 1 satisfies:* $\|\boldsymbol{w}_t^{(k)} - \boldsymbol{w}_t^{(1)}\|_\infty \leq AF$ *for all* $k \in \{1, \ldots, K\}$ *and* $t \in \{0, \ldots, T-1\}$.

Based on this Proposition, we can then prove the following convergence result.

**Theorem 3.9.** *Set*

$$\alpha = \min\left\{ \sqrt[3]{\frac{\Delta}{32KA^2\sigma^2 F_1^2 T}}, \frac{1}{26(K + A)(F + F_1)} \right\}.$$

*where $\sigma^2 = \max_m \sigma_m^2$ with $\sigma_m^2$ defined in Assumption 3.5. Under Assumptions 3.3, 3.4 and 3.5, Algorithm 1 yields*

$$\frac{1}{T} \sum_{t=0}^{T-1} \mathbb{E}\left[\min_{\lambda \in \Delta^M} \|\nabla \mathcal{L}(\boldsymbol{w}_t^{(1)})\lambda\|^2\right] \leq 11 \sqrt[3]{\frac{A^2 F_1^2 \Delta^2 (F^2 + \sigma^2)}{K^2 T^2}} + \frac{\sigma^2}{T} + \frac{65\Delta(F + F_1)}{T} + \frac{8\Delta A F_1}{KT}.$$

Proof is in Appendix A.3. Compared to random ordering in Theorem 3.6, the convergence rate of Algorithm 1 has a different term $\mathcal{O}((KT)^{-2/3})$ on the right hand side, which is better than the $\mathcal{O}((KT)^{-1/2})$ term in Theorem 3.6. As such, Algorithm 1 can achieve faster convergence than random ordering as is implemented in existing multi-objective optimization methods. We also note that a smaller $A$ leads to faster convergence, which demonstrates that solving the online vector balancing problem (minimizing $A$) is indeed useful to find better orders on the training samples.

The naive extension of GraB (Lu et al., 2022) that performs online vector balancing for gradients of each objective separately can also be analyzed under the same framework as follows:

**Proposition 3.10** (Separate Ordering). *Under Assumption 3.3 and 3.7, suppose that the sample order $\pi_t^m$ in Algorithm 1 is separately generated for each objective. We have $\|\boldsymbol{w}_t^{(k)} - \boldsymbol{w}_t^{(1)}\|_\infty \leq MAF$ for all $k \in \{1, \ldots, K\}$ and $t \in \{0, \ldots, T-1\}$.*

Proof is in Appendix A.4. Compared to the results in Proposition 3.8, the bound here is $M$ times larger if we apply gradient balancing separately on each objective. Recall that $M$ is the total number of objectives. Thus, the convergence can be much slower than that in Theorem 3.9.

## 4 EXPERIMENTS

In this section, we demonstrate the effectiveness of the proposed method for multi-objective optimization. We consider the following baselines: (i) Random reshuffling (Random), which is used in most existing implementations to randomly shuffle the whole data set in each epoch $t$, (ii) FlipFlop, which creates a new order $\pi_{t+1}$ by reversing the previous order, i.e., $\pi_{t+1}(k) = \pi_t(K + 1 - k)$. (iii) Random FlipFlop, the combination of random reshuffling and FlipFlop, and (iv) GraB (Lu et al.,

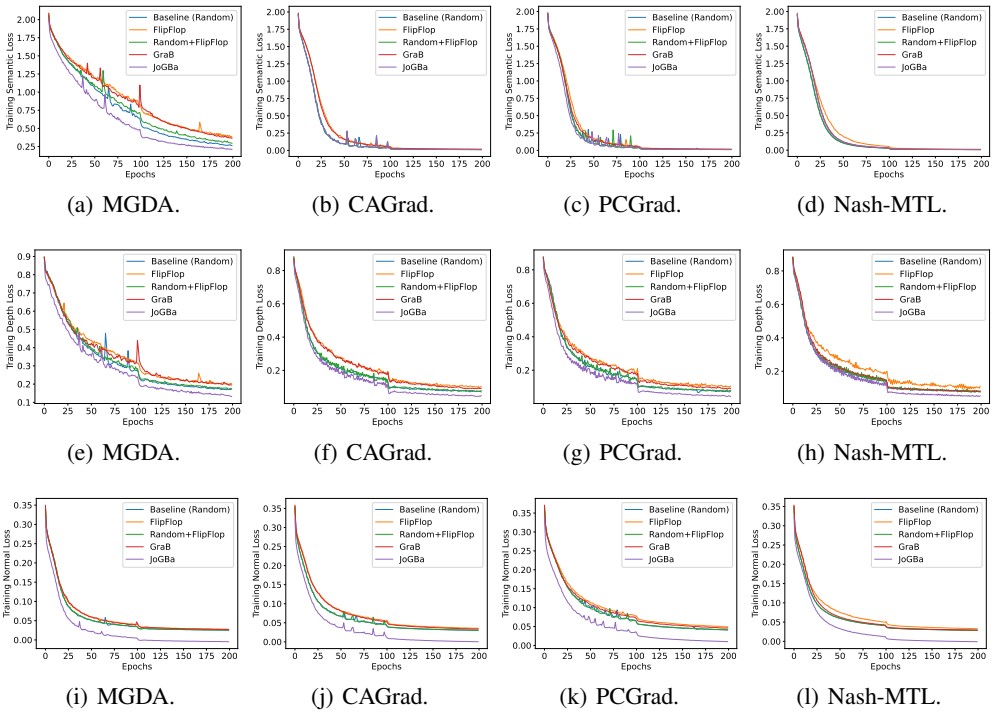

Figure 2: Training losses (objective values) of different tasks on NYUv2 data with different data ordering methods. Top: Loss on semantic segmentation task (semantic loss). Middle: Loss on depth estimation task (depth loss). Bottom: Loss on surface normal prediction task (normal loss).

2022), which performs gradient balancing on the weighted gradient of all objectives, and the weight is computed using the combined dynamic weighting algorithm.

The proposed method is independent of the dynamic weighting algorithms. In the following, we combine it with a variety of dynamic weighting algorithms, including: (i) MGDA (Désidéri, 2012; Liu & Vicente, 2021; Zhou et al., 2022; Fernando et al., 2023), (ii) PCGrad (Yu et al., 2020), (iii) CAGrad (Liu et al., 2021), and (iv) Nash-MTL (Navon et al., 2022). We select these methods as they generally have good empirical performance. The proposed method can also be easily combined with other dynamic weighting algorithms. Moreover, we include the single-task learning (Liu et al., 2021; Navon et al., 2022) baseline (STL), which trains on one task only and serves a performance upper bound.

We consider two data sets that are commonly used for multi-objective optimization in machine learning: (i) NYUv2 (Silberman et al., 2012), an indoor scene data set that involves three different tasks: semantic segmentation, depth estimation, and surface normal prediction. (ii) QM9 (Ramakrishnan et al., 2014), which is a widely used benchmark for graph neural networks predicting 11 properties of molecules. More details on the setup can be found in Appendix B.

## 4.1 NYUv2

Figure 2 compares the convergence curves of different ordering methods with the proposed method. We can see that the impact of sample order on the convergence can be different for different objectives. Both depth estimation and surface normal prediction tasks are more influenced by different sample ordering methods, while such influence becomes less significant for the semantic segmentation task. FlipFlop and GraB generally achieve worse performance than the other methods, while the proposed JoGBa is the only method that consistently outperforms existing baselines with random ordering.

Table 1 compares the testing performance of different data ordering combined with different multi-objective optimization methods. FlipFlop generally performs worse than the other methods as it only reverses the sample ordering after each epoch. Random FlipFlop slightly improves upon the standard random baseline. While GraB does not yield faster convergence rate in Figure 2, its testing

Table 1: Test performance (averaged over 3 random seeds) for the three tasks on NYUv2 data: semantic segmentation, depth estimation, and surface normal prediction.

| | Segmentation | | Depth | | Surface Normal | | | | | $\Delta \mathbf{m}\% \downarrow$ |
| --- | --- | --- | --- | --- | --- | --- | --- | --- | --- | --- |
| | | | | | Angle Distance ↓ | | Within $t°$ ↑ | | | |
| | mIoU ↑ | Pix Acc ↑ | Abs Err ↓ | Rel Err ↓ | Mean | Median | 11.25 | 22.5 | 30 | |
| STL | 38.30 | 63.76 | 0.6754 | 0.2780 | 25.01 | 19.21 | 30.14 | 57.20 | 69.15 | |
| MGDA (Random) | 30.48 | 59.77 | 0.6020 | 0.2555 | 24.13 | 19.22 | 29.51 | 57.11 | 69.58 | 1.31 |
| MGDA+FlipFlop | 29.47 | 57.90 | 0.6270 | 0.2755 | 24.88 | 19.45 | 29.18 | 55.88 | 68.36 | 1.58 |
| MGDA+Random FlipFlop | 30.52 | 59.81 | 0.6018 | 0.2556 | 24.11 | 19.16 | 29.52 | 57.23 | 69.56 | 1.28 |
| MGDA+GraB | 30.74 | 59.92 | 0.6011 | 0.2524 | 24.12 | 19.11 | 29.54 | 57.35 | 69.76 | 1.25 |
| MGDA+JoGBa | 31.02 | 60.21 | 0.6008 | 0.2508 | 24.08 | 19.08 | 29.55 | 57.47 | 70.03 | **1.19** |
| PCGrad (Random) | 38.06 | 64.64 | 0.5550 | 0.2325 | 27.41 | 22.80 | 23.86 | 49.83 | 63.14 | 3.97 |
| PCGrad+FlipFlop | 37.74 | 64.63 | 0.5590 | 0.2285 | 26.84 | 22.19 | 23.96 | 49.30 | 62.94 | 3.89 |
| PCGrad+Random FlipFlop | 38.12 | 64.64 | 0.5570 | 0.2329 | 26.99 | 22.67 | 23.56 | 49.65 | 63.18 | 3.86 |
| PCGrad+GraB | 38.31 | 64.66 | 0.5552 | 0.2317 | 26.79 | 22.87 | 23.68 | 49.76 | 63.22 | 3.78 |
| PCGrad+JoGBa | 38.59 | 64.67 | 0.5545 | 0.2270 | 26.53 | 22.40 | 23.87 | 49.95 | 63.87 | **3.56** |
| CAGrad (Random) | 39.79 | 65.49 | 0.5486 | 0.2250 | 26.31 | 21.58 | 25.61 | 52.36 | 65.58 | 0.20 |
| CAGrad+FlipFlop | 39.42 | 65.55 | 0.5437 | 0.2219 | 25.79 | 21.75 | 25.97 | 52.17 | 65.34 | 0.27 |
| CAGrad+Random FlipFlop | 39.85 | 65.73 | 0.5467 | 0.2226 | 26.14 | 21.46 | 25.62 | 52.24 | 65.62 | 0.17 |
| CAGrad+GraB | 39.91 | 66.09 | 0.5428 | 0.2214 | 25.79 | 21.44 | 25.64 | 52.26 | 65.44 | 0.18 |
| CAGrad+JoGBa | 40.42 | 66.08 | 0.5410 | 0.2205 | 25.52 | 21.50 | 26.04 | 52.43 | 65.73 | **0.03** |
| Nash-MTL (Random) | 40.13 | 65.93 | 0.5261 | 0.2171 | 25.26 | 20.08 | 28.40 | 55.47 | 68.15 | −4.04 |
| Nash-MTL+FlipFlop | 39.46 | 65.82 | 0.5313 | 0.2190 | 26.12 | 20.99 | 28.05 | 54.64 | 67.77 | -3.88 |
| Nash-MTL+Random FlipFlop | 40.67 | 66.32 | 0.5184 | 0.2009 | 25.34 | 19.73 | 28.54 | 55.35 | 68.07 | -4.16 |
| Nash-MTL+GraB | 40.84 | 66.51 | 0.5156 | 0.2087 | 25.26 | 19.45 | 28.62 | 55.37 | 68.11 | -4.19 |
| Nash-MTL+JoGBa | 41.13 | 66.71 | 0.5112 | 0.2009 | 25.11 | 19.19 | 28.77 | 55.28 | 68.18 | **-4.27** |

performance is comparable to Random FlipFlop. The proposed method JoGBa achieves the best overall performance across different performance metrics for all three tasks.

## 4.2 QM9

Due to the large number of objectives in the QM9 data, here we only plot the average of all training objectives. The convergence curves are shown in Figure 3 for different sample ordering methods. Compared to the NYUv2 data set, the effect of sample ordering becomes less significant for the QM9 data. Only GraB and JoGBa achieve slight improvements than the other ordering methods.

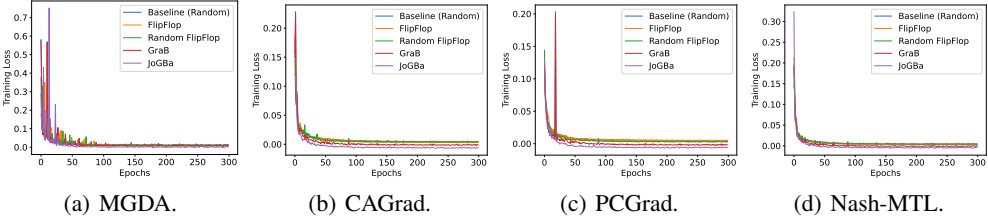

(a) MGDA.          (b) CAGrad.          (c) PCGrad.          (d) Nash-MTL.

Figure 3: Training losses on QM9 data with different data ordering methods.

Table 2 compares the testing performance of different data ordering methods. Similar to the results for NYUv2, FlipFlop generally performs worse as it only reverses the sample ordering after each epoch. Random FlipFlop achieves comparable performance with the random ordering baseline, and GraB slightly improves upon it. The proposed method JoGBa achieves the best overall performance.

## 4.3 COMPARISON ON TIME COSTS

The proposed JoGBa has two key steps in each iteration: (i) sample ordering, in which we determine the order of a sample based on its gradients, and (ii) model updating, in which we compute the objective weights and update the model with the weighted gradients. Table 3 compares the time costs of these two steps in each iteration for different multi-objective optimization algorithms on NYUv2 and QM9 data. As can be seen, the time cost of sample ordering is almost negligible compared to that of model update, and is generally the same for the same data set across different multi-objective optimization algorithms. This is intuitive as sample ordering is not related to any specific multi-objective optimization algorithm, and demonstrates that the proposed method does not introduce much additional time cost.

Table 2: Test performance (averaged over 3 random seeds) on all property prediction tasks in QM9.

| | $\mu$ | $\alpha$ | $\epsilon_{\text{HOMO}}$ | $\epsilon_{\text{LUMO}}$ | $\langle R^2 \rangle$ | ZPVE | $U_0$ | $U$ | $H$ | $G$ | $c_v$ | |
|---|---|---|---|---|---|---|---|---|---|---|---|---|
| | | | | | | MAE ↓ | | | | | | $\Delta_{\mathbf{m}}\%\downarrow$ |
| STL | 0.067 | 0.181 | 60.57 | 53.91 | 0.502 | 4.53 | 58.8 | 64.2 | 63.8 | 66.2 | 0.072 | |
| MGDA (Random) | 0.217 | 0.368 | 126.8 | 104.6 | 3.22 | 5.69 | 88.37 | 89.40 | 89.32 | 88.01 | 0.120 | 120.5 |
| MGDA+FlipFlop | 0.221 | 0.371 | 130.9 | 104.5 | 3.32 | 5.62 | 88.31 | 89.45 | 89.71 | 88.84 | 0.124 | 121.4 |
| MGDA+Random FlipFlop | 0.216 | 0.365 | 126.7 | 103.2 | 3.19 | 5.65 | 88.34 | 89.27 | 88.74 | 87.34 | 0.115 | 118.9 |
| MGDA+GraB | 0.206 | 0.343 | 120.8 | 101.4 | 3.16 | 5.44 | 87.68 | 88.63 | 88.87 | 87.26 | 0.119 | 118.4 |
| MGDA+JoGBa | 0.202 | 0.332 | 117.3 | 99.2 | 3.12 | 5.37 | 87.48 | 88.37 | 88.80 | 87.04 | 0.116 | **116.7** |
| PCGrad (Random) | 0.106 | 0.293 | 75.85 | 88.33 | 3.94 | 9.15 | 116.36 | 116.8 | 117.2 | 114.5 | 0.110 | 125.7 |
| PCGrad+FlipFlop | 0.106 | 0.306 | 75.15 | 88.29 | 3.87 | 9.17 | 120.17 | 117.4 | 117.8 | 114.1 | 0.113 | 126.3 |
| PCGrad+Random FlipFlop | 0.104 | 0.293 | 75.05 | 88.25 | 3.83 | 9.07 | 114.89 | 116.4 | 116.9 | 114.1 | 0.106 | 125.2 |
| PCGrad+GraB | 0.098 | 0.281 | 74.91 | 86.98 | 3.75 | 8.91 | 115.66 | 114.4 | 117.1 | 113.6 | 0.102 | 124.2 |
| PCGrad+JoGBa | 0.098 | 0.271 | 74.43 | 84.30 | 3.56 | 8.78 | 113.15 | 113.2 | 117.1 | 113.5 | 0.096 | **123.5** |
| CAGrad (Random) | 0.118 | 0.321 | 83.51 | 94.81 | 3.21 | 6.93 | 113.99 | 114.3 | 114.5 | 112.3 | 0.116 | 112.8 |
| CAGrad+FlipFlop | 0.115 | 0.325 | 85.13 | 94.94 | 3.24 | 7.09 | 114.32 | 115.2 | 114.9 | 113.1 | 0.117 | 113.1 |
| CAGrad+Random FlipFlop | 0.113 | 0.322 | 83.19 | 94.87 | 3.15 | 6.92 | 114.18 | 113.8 | 113.8 | 111.6 | 0.113 | 112.8 |
| CAGrad+GraB | 0.111 | 0.312 | 82.49 | 94.71 | 2.96 | 6.77 | 113.89 | 113.7 | 110.4 | 111.8 | 0.108 | 112.1 |
| CAGrad+JoGBa | 0.110 | 0.304 | 82.38 | 94.49 | 2.92 | 6.49 | 113.22 | 113.5 | 110.2 | 111.6 | 0.104 | **111.9** |
| Nash-MTL (Random) | 0.102 | 0.248 | 82.95 | 81.89 | 2.42 | 5.38 | 74.50 | 75.02 | 75.10 | 74.16 | 0.093 | 62.0 |
| Nash-MTL+FlipFlop | 0.106 | 0.255 | 82.79 | 82.01 | 2.45 | 5.42 | 74.52 | 75.07 | 75.13 | 74.27 | 0.096 | 62.2 |
| Nash-MTL+Random FlipFlop | 0.097 | 0.254 | 82.53 | 81.47 | 2.42 | 5.29 | 74.41 | 75.08 | 75.07 | 74.22 | 0.094 | 61.6 |
| Nash-MTL+GraB | 0.099 | 0.252 | 82.64 | 81.68 | 2.38 | 5.31 | 74.43 | 74.94 | 75.05 | 74.13 | 0.091 | 61.7 |
| Nash-MTL+JoGBa | 0.094 | 0.231 | 82.24 | 80.73 | 2.29 | 5.24 | 74.37 | 74.84 | 75.03 | 74.05 | 0.087 | **59.2** |

Table 3: Per-iteration CPU time cost (in seconds) of the two key steps in JoGBa combined with different multi-objective optimization algorithms.

| | NYUv2 | | | | QM9 | | | |
|---|---|---|---|---|---|---|---|---|
| | MGDA | PCGrad | CAGrad | Nash-MTL | MGDA | PCGrad | CAGrad | Nash-MTL |
| Sample ordering | 0.02 | 0.03 | 0.03 | 0.03 | 0.06 | 0.05 | 0.04 | 0.05 |
| Model update | 1.04 | 0.91 | 0.99 | 1.06 | 2.97 | 1.37 | 1.17 | 1.62 |

Table 4: Test performance (averaged over 3 random seeds) for three tasks on NYUv2 with different sample ordering methods for the proposed multi-ordering framework.

| | Segmentation | | Depth | | Surface Normal | | | | | |
|---|---|---|---|---|---|---|---|---|---|---|
| | | | | | Angle Distance ↓ | | Within $t°$ ↑ | | | $\Delta\mathbf{m}\%\downarrow$ |
| | mIoU ↑ | Pix Acc ↑ | Abs Err ↓ | Rel Err ↓ | Mean | Median | 11.25 | 22.5 | 30 | |
| MGDA+Random | 30.48 | 59.77 | 0.6020 | 0.2555 | 24.13 | 19.22 | 29.51 | 57.11 | 69.58 | 1.31 |
| MGDA+FlipFlop | 29.47 | 57.90 | 0.6270 | 0.2755 | 24.88 | 19.45 | 29.18 | 55.88 | 68.36 | 1.58 |
| MGDA+Random FlipFlop | 30.52 | 59.81 | 0.6018 | 0.2556 | 24.11 | 19.16 | 29.52 | 57.23 | 69.56 | 1.28 |
| MGDA+GraB | 30.74 | 59.92 | 0.6011 | 0.2524 | 24.12 | 19.11 | 29.54 | 57.35 | 69.76 | 1.25 |
| MGDA+JoGBa | 31.02 | 60.21 | 0.6008 | 0.2508 | 24.08 | 19.08 | 29.55 | 57.47 | 70.03 | **1.19** |
| PCGrad+Random | 38.06 | 64.64 | 0.5550 | 0.2325 | 27.41 | 22.80 | 23.86 | 49.83 | 63.14 | 3.97 |
| PCGrad+FlipFlop | 37.74 | 64.63 | 0.5590 | 0.2285 | 26.84 | 22.19 | 23.96 | 49.30 | 62.94 | 3.89 |
| PCGrad+Random FlipFlop | 38.12 | 64.64 | 0.5570 | 0.2329 | 26.99 | 22.67 | 23.56 | 49.65 | 63.18 | 3.86 |
| PCGrad+GraB | 38.31 | 64.66 | 0.5552 | 0.2317 | 26.79 | 22.87 | 23.68 | 49.76 | 63.22 | 3.78 |
| PCGrad+JoGBa | 38.59 | 64.67 | 0.5545 | 0.2270 | 26.53 | 22.40 | 23.87 | 49.95 | 63.87 | **3.56** |
| CAGrad+Random | 39.79 | 65.49 | 0.5486 | 0.2250 | 26.31 | 21.58 | 25.61 | 52.36 | 65.58 | 0.20 |
| CAGrad+FlipFlop | 39.42 | 65.55 | 0.5437 | 0.2219 | 25.79 | 21.75 | 25.97 | 52.17 | 65.34 | 0.27 |
| CAGrad+Random FlipFlop | 39.85 | 65.73 | 0.5467 | 0.2226 | 26.14 | 21.46 | 25.62 | 52.24 | 65.62 | 0.17 |
| CAGrad+GraB | 39.91 | 66.09 | 0.5428 | 0.2214 | 25.79 | 21.44 | 25.64 | 52.26 | 65.44 | 0.18 |
| CAGrad+JoGBa | 40.42 | 66.08 | 0.5410 | 0.2205 | 25.52 | 21.50 | 26.04 | 52.43 | 65.73 | **0.03** |
| Nash-MTL+Random | 40.13 | 65.93 | 0.5261 | 0.2171 | 25.26 | 20.08 | 28.40 | 55.47 | 68.15 | −4.04 |
| Nash-MTL+FlipFlop | 39.46 | 65.82 | 0.5313 | 0.2190 | 26.12 | 20.99 | 28.05 | 54.64 | 67.77 | -3.88 |
| Nash-MTL+Random FlipFlop | 40.67 | 66.32 | 0.5184 | 0.2009 | 25.34 | 19.73 | 28.54 | 55.35 | 68.07 | -4.16 |
| Nash-MTL+GraB | 40.84 | 66.51 | 0.5156 | 0.2087 | 25.26 | 19.45 | 28.62 | 55.37 | 68.11 | -4.19 |
| Nash-MTL+JoGBa | 41.13 | 66.71 | 0.5112 | 0.2009 | 25.11 | 19.19 | 28.77 | 55.28 | 68.18 | **-4.27** |

## 4.4 ABLATION STUDY

To solve the underlying online vector balancing balancing problem, besides using Algorithm 1, other data ordering methods (such as those mentioned in Section 3.1) may also be used to obtain

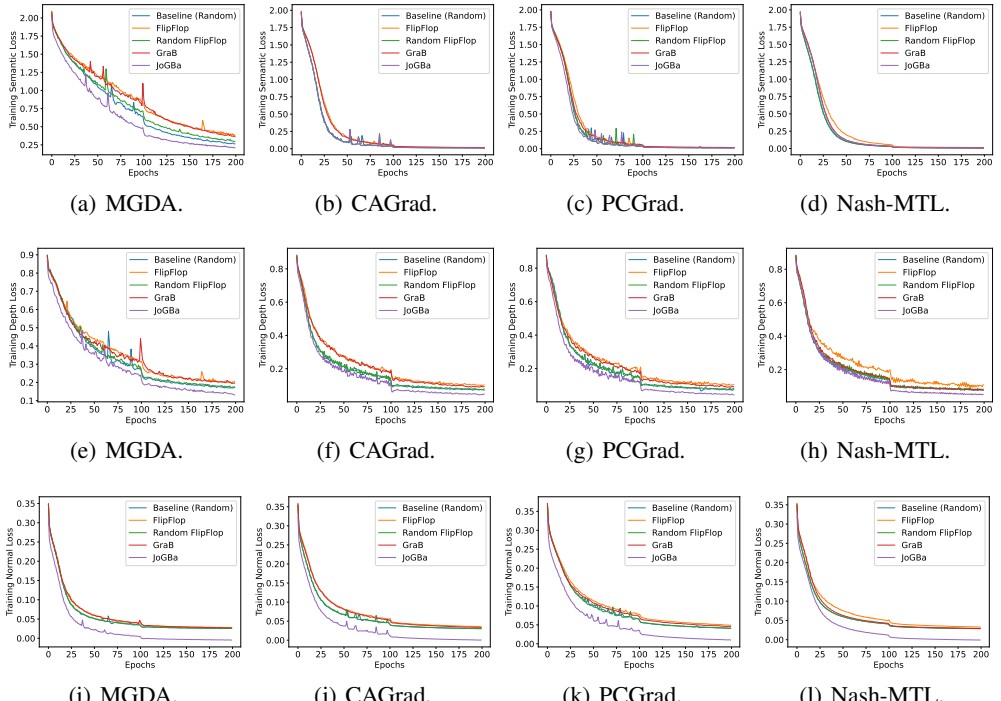

Figure 4: Training loss (objective values) of different tasks on NYUv2 data with different data ordering methods for the proposed multi-ordering framework. Top: Loss on the semantic segmentation task (semantic loss). Middle: Loss on the depth estimation task (depth loss). Bottom: Loss on the surface normal prediction task (normal loss).

sample orders for different objectives. In this experiment, we consider the following sample ordering methods for comparison: (i) Random reshuffling (Random), (ii) FlipFlop, which creates the new order $\pi_{t+1}^m$ by reversing the previous order for each objective, i.e., $\pi_{t+1}^m(k) = \pi_t^m(K + 1 - k)$. (iii) Random FlipFlop (Random FF), the combination of random reshuffling and FlipFlop, and (iv) GraB (Lu et al., 2022), which applies GraB to all objectives separately. Experiment is performed on the same NYUv2 data set and training setup as in Section 4.1.

Figure 4 compares the convergence curves of different sample ordering methods. Similar to Figure 2, the influence of sample orders on the convergence rate is generally different for different objectives. The surface normal prediction task is more influenced by different sample ordering methods than the other two tasks. FlipFlop and GraB generally achieves worse performance than the other methods, while JoGBa is the only method that consistently outperforms existing baselines.

Table 4 compares the testing performance of different data ordering combined with different multi-objective optimization methods. FlipFlop generally performs worse than other methods as it only reverses the sample ordering after each epoch. Both Random FlipFlop and GraB improve upon the standard random baseline, but their performance is still worse than the proposed method JoGBa, which demonstrate the effectiveness of joint sample ordering in multi-objective optimization.

## 5 CONCLUSION

In this paper, we propose a novel sample ordering framework for multi-objective optimization. The proposed framework determines sample orders for each objective by performing online vector balancing with the gradients on different objectives. It can be seamlessly combined with existing multi-objective optimization methods. Theoretical results demonstrate that the proposed method improves upon the baseline of random ordering with faster convergence. Empirical results on different multi-objective optimization problems demonstrate that the proposed method achieves faster convergence and better performance than the other data ordering methods.

ACKNOWLEDGMENTS

This research is supported in part by the Research Grants Council of the Hong Kong Special Administrative Region (Grants 16200021, 16202523 and C7004-22G-1).

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

# A PROOFS

## A.1 PROOF OF THEOREM 3.6

*Theorem 3.6.* By the $F_1$-smoothness of $\mathcal{L}(\boldsymbol{w})\lambda$ for all $\lambda \in \Delta^M$, we have

$$\mathcal{L}(\boldsymbol{w}_{t+1})\lambda - \mathcal{L}(\boldsymbol{w}_t)\lambda \leq \langle \nabla \mathcal{L}(\boldsymbol{w})\lambda, \boldsymbol{w}_{t+1} - \boldsymbol{w}_t \rangle + \frac{F_1}{2}\|\boldsymbol{w}_{t+1} - \boldsymbol{w}_t\|^2 \tag{3}$$

where $\boldsymbol{w}_{t+1} - \boldsymbol{w}_t = \alpha_t \nabla \mathcal{L}(\boldsymbol{w}_t)\lambda_t^*$, s.t. $\lambda_t^* \in \arg\min_{\lambda \in \Delta^M} \|\nabla \mathcal{L}(\boldsymbol{w}_t)\lambda\|^2$. For notation simplicity, we define $Q_t = \nabla \mathcal{L}(\boldsymbol{w}_t)$, and $\lambda_{Q_t}^* = \arg\min_{\lambda \in \Delta^M} \|\nabla \mathcal{L}(\boldsymbol{w}_t)\lambda\|$. Then we have:

$$\mathcal{L}(\boldsymbol{w}_{t+1})\lambda - \mathcal{L}(\boldsymbol{w}_t)\lambda \leq -\alpha_t \langle \nabla \mathcal{L}(\boldsymbol{w}_t)\lambda, Q_t\lambda_{Q_t}^* \rangle + \frac{F_1}{2}\alpha_t^2 \|Q_t\lambda_{Q_t}^*\|^2. \tag{4}$$

The inner product term can be bounded as

$$
\begin{aligned}
-\langle \nabla \mathcal{L}(\boldsymbol{w}_t)\lambda, Q_t\lambda_{Q_t}^* \rangle &= \langle \nabla \mathcal{L}(\boldsymbol{w}_t)\lambda, \nabla \mathcal{L}(\boldsymbol{w}_t)\lambda_t^*(\boldsymbol{w}_t) - Q_t\lambda_{Q_t}^* \rangle - \langle \nabla \mathcal{L}(\boldsymbol{w}_t)\lambda, \nabla \mathcal{L}(\boldsymbol{w}_t)\lambda_t^*(\boldsymbol{w}_t) \rangle \\
&\overset{(a)}{\leq} \langle \nabla \mathcal{L}(\boldsymbol{w}_t)\lambda, \nabla \mathcal{L}(\boldsymbol{w}_t)\lambda_t^*(\boldsymbol{w}_t) - Q_t\lambda_{Q_t}^* \rangle - \|\nabla \mathcal{L}(\boldsymbol{w}_t)\lambda_t^*(\boldsymbol{w}_t)\|^2 \\
&\leq F\|\nabla \mathcal{L}(\boldsymbol{w}_t)\lambda_t^*(\boldsymbol{w}_t) - Q_t\lambda_{Q_t}^*\| - \|\nabla \mathcal{L}(\boldsymbol{w}_t)\lambda_t^*(\boldsymbol{w}_t)\|^2 \\
&\overset{(b)}{\leq} 2F^{\frac{3}{2}}\|Q_t - \nabla \mathcal{L}(\boldsymbol{w}_t)\|^{\frac{1}{2}} - \|\nabla \mathcal{L}(\boldsymbol{w}_t)\lambda_t^*(\boldsymbol{w}_t)\|^2
\end{aligned}
\tag{5}$$

where $(a)$ follows from (13) in Lemma A.3, $(b)$ follows from Lemma A.4. Plugging (5) into (4), taking expectations on both sides and rearranging yields

$$
\begin{aligned}
\alpha_t \mathbb{E}_A[\|\nabla \mathcal{L}(\boldsymbol{w}_t)\lambda_t^*(\boldsymbol{w}_t)\|^2] \leq{}& \mathbb{E}_A[\mathcal{L}(\boldsymbol{w}_t) - \mathcal{L}(\boldsymbol{w}_{t+1})]\lambda + 2F^{\frac{3}{2}}\alpha_t \mathbb{E}_A[\|Q_t - \nabla \mathcal{L}(\boldsymbol{w}_t)\|^{\frac{1}{2}}] \\
&+ \frac{F_1}{2}(F^2 + \sigma^2)\alpha_t^2.
\end{aligned}
$$

For all $t \in [T]$, plugging in $\alpha_t = \alpha$, and taking the telescope sum on both sides of the last inequality yield

$$
\begin{aligned}
\frac{1}{T}\sum_{t=1}^{T}& \mathbb{E}_A[\|\nabla \mathcal{L}(\boldsymbol{w}_t)\lambda_t^*(x_t)\|^2] \\
&\leq \frac{1}{\alpha T}\mathbb{E}_A[\mathcal{L}(\boldsymbol{w}_t) - \mathcal{L}(\boldsymbol{w}_{t+1})]\lambda + 2\ell_f^{\frac{3}{2}}\frac{1}{T}\sum_{t=1}^{T}\mathbb{E}_A[\|Q_t - \nabla \mathcal{L}(\boldsymbol{w}_t)\|^{\frac{1}{2}}] + \frac{F_1}{2}(F^2 + \sigma^2)\alpha \\
&\leq \frac{1}{\alpha T}\mathbb{E}_A[\mathcal{L}(\boldsymbol{w}_t) - \mathcal{L}(\boldsymbol{w}_{t+1})]\lambda + 2\ell_f^{\frac{3}{2}}\left(\frac{1}{T}\sum_{t=1}^{T}\mathbb{E}_A[\|Q_t - \nabla \mathcal{L}(\boldsymbol{w}_t)\|^2]\right)^{\frac{1}{4}} + \frac{F_1}{2}(F^2 + \sigma^2)\alpha.
\end{aligned}
\tag{6}$$

By increasing the batch size during optimization with a batch size of $\mathcal{O}(t)$, it holds that

$$\frac{1}{T}\sum_{t=1}^{T}\mathbb{E}_A[\|Q_t - \nabla \mathcal{L}(\boldsymbol{w}_t)\|^2] \leq \frac{1}{T}\sum_{t=1}^{T}\frac{\sigma^2}{t} \leq \frac{\sigma^2(1 + \log(T))}{T}. \tag{7}$$

Plugging (7) back into (6), its optimization error is given by:

$$
\begin{aligned}
\mathbb{E}_A\Big[\min_{t \in [T], \lambda \in \Delta^M} \|\nabla \mathcal{L}(\boldsymbol{w}_t)\lambda\|^2\Big] &\leq \frac{1}{T}\sum_{t=1}^{T}\mathbb{E}_A[\|\nabla \mathcal{L}(\boldsymbol{w}_t)\lambda_t^*(\boldsymbol{w}_t)\|^2] \\
&= \frac{\sigma^2(1 + \log(T))}{T} + \frac{\mathbb{E}_A[\mathcal{L}(\boldsymbol{w}_t) - \mathcal{L}(\boldsymbol{w}_{T+1})]\lambda}{\alpha K T} + \frac{F_1}{2}(F^2 + \sigma^2)\alpha \\
&\leq \frac{\sigma^2(1 + \log(T))}{T} + \frac{\Delta}{\alpha K T} + \frac{F_1}{2}(F^2 + \sigma^2)\alpha,
\end{aligned}
\tag{8}$$

where the last inequality uses the definition of $\Delta = \max_{\lambda \in \Delta^M} \mathcal{L}(\boldsymbol{w}_0)\lambda - \min_{\boldsymbol{w} \in \mathbb{R}^d, \lambda \in \Delta^M} \mathcal{L}(\boldsymbol{w})\lambda$. Then setting $\alpha = \sqrt{\frac{2\Delta}{F_1(F^2 + \sigma^2)KT}}$, we have:

$$\mathbb{E}_A\Big[\min_{t \in [T], \lambda \in \Delta^M} \|\nabla \mathcal{L}(\boldsymbol{w}_t)\lambda\|^2\Big] \leq \sqrt{\frac{2F_1\Delta(F^2 + \sigma^2)}{KT}} + \frac{\sigma^2(1 + \log(T))}{T},$$

which concludes our proof. $\qquad\square$

## A.2 PROOF OF PROPOSITION 3.8

*Proof.* The proof follows directly by using Assumption 3.3 in Assumption 3.7 for each $\|\nabla\ell_i(\boldsymbol{w})\|_\infty \leq F$. $\qquad\square$

## A.3 PROOF OF THEOREM 3.9

*Proof.* From Lemma A.1 in Appendix A.5, we have

$$\frac{1}{T}\sum_{t=0}^{T-1}\min_{\lambda\in\Delta^M}\|\nabla\mathcal{L}(\boldsymbol{w}_t^{(1)})\lambda\|^2 \leq \frac{2\Delta}{\alpha KT} + \frac{2F_1^2}{T}\sum_{t=0}^{T-1}\max_k\left\|\boldsymbol{w}_t^{(k)}-\boldsymbol{w}_t^{(1)}\right\|_\infty^2 + \frac{\alpha^2 F_1(F^2+\sigma^2)}{2}.$$

On the other hand, from Lemma A.2, we obtain

$$\sum_{t=0}^{T-1}\Delta_t^2 \leq 120\alpha^2 K^2\sigma^2 + 64\alpha^2 A^2\sigma^2 T + 48\alpha^2 K^2\sum_{t=0}^{T-1}\max_k\|\nabla\mathcal{L}(\boldsymbol{w}_t^{(k)})\lambda\|_\infty^2.$$

Combining them together gives:

$$\frac{1}{T}\sum_{t=0}^{T-1}\min_{\lambda\in\Delta^M}\|\nabla\mathcal{L}(\boldsymbol{w}_t^{(1)})\lambda\|^2$$

$$\leq \quad \frac{2\Delta}{\alpha KT} + \frac{F_1^2}{T}\left(120\alpha^2 K^2\sigma^2 + 64\alpha^2 A^2\sigma^2 T + 48\alpha^2 K^2\sum_{t=0}^{T-1}\max_k\|\nabla\mathcal{L}(\boldsymbol{w}_t^{(k)})\lambda\|_\infty^2\right)$$

$$+\frac{\alpha^2 F_1(F^2+\sigma^2)}{2}$$

$$\leq \quad \frac{2\Delta}{\alpha KT} + \frac{120\alpha^2 F_1^2 K^2\sigma^2}{T} + 64\alpha^2 A^2 F_1^2\sigma^2 + \frac{48\alpha^2 K^2 F_1^2}{T}\sum_{t=0}^{T-1}\max_k\|\nabla\mathcal{L}(\boldsymbol{w}_t^{(k)})\lambda\|_\infty^2$$

$$+\frac{\alpha^2 F_1(F^2+\sigma^2)}{2}.$$

Note that for any $\boldsymbol{x}\in\mathbb{R}^d$, $\|\boldsymbol{x}\|_\infty \leq \|\boldsymbol{x}\|_2$, so the last term can by bounded by its $\ell_2$-norm. Moving it to the left side of the inequality gives:

$$\frac{1-48\alpha^2 K^2 F_1^2}{T}\sum_{t=0}^{T-1}\min_{\lambda\in\Delta^M}\|\nabla\mathcal{L}(\boldsymbol{w}_t^{(1)})\lambda\|^2 \leq \frac{2\Delta}{\alpha KT} + \frac{120\alpha^2 F_1^2 K^2\sigma^2}{T} + 64\alpha^2 A^2 F_1^2\sigma^2$$

$$+\frac{\alpha^2 F_1(F^2+\sigma^2)}{2}.$$

Finally, we set the value of $\alpha$ as:

$$\alpha = \min\left\{\sqrt[3]{\frac{\Delta}{32KA^2\sigma^2 F_1^2 T}}, \frac{1}{KF}, \frac{1}{26(K+A)F_1}\right\},$$

and we obtain

$$\frac{1}{T}\sum_{t=0}^{T-1}\min_{\lambda\in\Delta^M}\|\nabla\mathcal{L}(\boldsymbol{w}_t^{(1)})\lambda\|^2 \leq 11\sqrt[3]{\frac{A^2 F_1^2\Delta^2(F^2+\sigma^2)}{K^2 T^2}} + \frac{\sigma^2}{T} + \frac{65\Delta(F+F_1)}{T} + \frac{8\Delta AF_1}{KT},$$

which concludes the proof. $\qquad\square$

## A.4 PROOF FOR PROPOSITION 3.10

*Proof.* Similar to Proposition 3.8, the proof follows directly by using Assumption 3.3 in Assumption 3.7 for each $\|\nabla\ell_i(\boldsymbol{w})\|_\infty \leq F$, and repeated for $M$ objectives. $\qquad\square$

## A.5 TECHNICAL LEMMAS

**Lemma A.1.** *In Algorithm 1, if $\alpha K F < 1$ holds and Assumption 3.3 and 3.4 hold, then*

$$\frac{1}{T} \sum_{t=0}^{T-1} \min_{\lambda \in \Delta^M} \|\nabla \mathcal{L}(\boldsymbol{w}_t^{(1)})\lambda\|^2 \leq \frac{2\Delta}{\alpha K T} + \frac{2F_1^2}{T} \sum_{t=0}^{T-1} \max_k \left\|\boldsymbol{w}_t^{(k)} - \boldsymbol{w}_t^{(1)}\right\|_\infty^2 + \frac{\alpha^2 F_1(F^2 + \sigma^2)}{2}.$$

*Proof.* Note that the update can be written as

$$\boldsymbol{w}_{t+1}^{(1)} = \boldsymbol{w}_t^{(1)} - \alpha \sum_{k=1}^K \sum_{m=1}^M \lambda_{k,m} \nabla \ell_m(\boldsymbol{w}_t^{(k)}; \xi_{\pi_k^m(t)}).$$

By Taylor's Theorem, for all $t = 0, \cdots, T-1$,

$$
\begin{aligned}
\mathcal{L}(\boldsymbol{w}_{t+1}^{(1)})\lambda &\leq \mathcal{L}(\boldsymbol{w}_t^{(1)})\lambda + \langle \nabla \mathcal{L}(x_t)\lambda, \boldsymbol{w}_{t+1}^{(1)} - \boldsymbol{w}_t^{(1)} \rangle + \frac{F_1}{2} \|\boldsymbol{w}_{t+1}^{(1)} - \boldsymbol{w}_t^{(1)}\|^2 \\
&\leq \mathcal{L}(\boldsymbol{w}_t^{(1)})\lambda - \alpha K \mathbb{E} \left\langle \nabla \mathcal{L}(\boldsymbol{w}_t)\lambda, \frac{1}{K} \sum_{k=1}^K \sum_{m=1}^M \lambda_{i,k} \nabla \ell_i(\boldsymbol{w}_t^{(k)}; \xi_{\pi_t^m(t)}) \right\rangle \\
&\quad + \frac{\alpha^2 K^2 F_1}{2} \mathbb{E} \left\| \frac{1}{K} \sum_{k=1}^K \sum_{m=1}^M \lambda_{i,k} \nabla \ell_i(\boldsymbol{w}_t^{(k)}; \xi_{\pi_i(t)}) \right\|^2 \\
&= \mathcal{L}(\boldsymbol{w}_t^{(1)})\lambda - \frac{\alpha K}{2} \|\nabla \mathcal{L}(\boldsymbol{w}_t^{(1)})\lambda\|^2 - \frac{\alpha K}{2} \| \frac{1}{K} \sum_{k=1}^K \sum_{m=1}^M \lambda_{i,k} \nabla \ell_i(\boldsymbol{w}_t^{(k)}; \xi_{\sigma_k(t)}) \|^2 \\
&\quad + \frac{\alpha K}{2} \| \nabla \mathcal{L}(\boldsymbol{w}_t^{(1)})\lambda - \frac{1}{K} \sum_{k=1}^K \sum_{m=1}^M \lambda_{i,k} \nabla \ell_i(\boldsymbol{w}_t^{(k)}; \xi_{\sigma_k(t)}) \|^2 \\
&\quad + \frac{\alpha^2 K^2 F_1}{2} \mathbb{E} \left\| \frac{1}{K} \sum_{k=1}^K \sum_{i=1}^m \lambda_{i,k} \nabla \ell_i(\boldsymbol{w}_k^{(t)}; \xi_{\sigma_k(t)}) \right\|^2 \\
&\leq \mathcal{L}(\boldsymbol{w}_t^{(1)})\lambda - \frac{\alpha K}{2} \|\nabla \mathcal{L}(\boldsymbol{w}_t^{(1)})\lambda\|^2 + \frac{\alpha K}{2} \|\nabla \mathcal{L}(\boldsymbol{w}_t^{(1)})\lambda - \frac{1}{K} \sum_{k=1}^K \sum_{m=1}^M \lambda_{i,k} \nabla \ell_i(\boldsymbol{w}_t^{(k)}; \xi_{\sigma_k(t)}) \|^2 \\
&\quad + \frac{\alpha^2 F_1(F^2 + \sigma^2)}{2}.
\end{aligned}
$$

In the second step, we apply $-\langle \boldsymbol{a}, \boldsymbol{b} \rangle = -\frac{1}{2}\|\boldsymbol{a}\|^2 - \frac{1}{2}\|\boldsymbol{b}\|^2 + \frac{1}{2}\|\boldsymbol{a} - \boldsymbol{b}\|^2, \forall \boldsymbol{a}, \boldsymbol{b}$. In the third step, we use the condition that $\alpha n L < 1$. Expanding the last term using Assumption 3.4, we get

$$
\begin{aligned}
\|\nabla \mathcal{L}(\boldsymbol{w}_t^{(1)})\lambda &- \frac{1}{K} \sum_{k=1}^K \sum_{m=1}^M \lambda_{m,k,t} \nabla \ell_m(\boldsymbol{w}_t^{(k)}; \xi_{\sigma_t^m(k)}) \|^2 \\
&= \left\| \frac{1}{K} \sum_{k=1}^K \nabla \mathcal{L}(\boldsymbol{w}_t^{(1)})\lambda - \frac{1}{K} \sum_{k=1}^K \sum_{m=1}^M \lambda_{m,k,t} \nabla \ell_m(\boldsymbol{w}_t^{(k)}; \xi_{\sigma_t^m(k)}) \right\|^2 \\
&\leq \frac{1}{K} \sum_{k=1}^K \left\| \nabla \mathcal{L}(\boldsymbol{w}_t^{(1)})\lambda - \nabla \mathcal{L}(\boldsymbol{w}_t^{(k)})\lambda \right\|^2 \\
&\leq \frac{1}{K} \sum_{k=1}^K F_1^2 \left\| \boldsymbol{w}_t^{(1)} - \boldsymbol{w}_t^{(k)} \right\|_\infty^2 \\
&\leq F_1^2 \Delta_k^2.
\end{aligned}
$$

In the second step we apply Jensen's Inequality. Putting it back, we obtain

$$\mathcal{L}(\boldsymbol{w}_{t+1}^{(1)})\lambda \leq \mathcal{L}(\boldsymbol{w}_t^{(1)})\lambda - \frac{\alpha K}{2} \left\| \nabla \mathcal{L}(\boldsymbol{w}_t^{(1)})\lambda \right\|^2 + \frac{\alpha K}{2} F_1^2 \Delta_k^2 + \frac{\alpha^2 F_1(F^2 + \sigma^2)}{2}.$$

Finally, summing from $t = 0$ to $T - 1$, and using the definition $\Delta = \max_{\lambda \in \Delta^M} \mathcal{L}(\boldsymbol{w}_0)\lambda - \min_{\boldsymbol{w} \in \mathbb{R}^d, \lambda \in \Delta^M} \mathcal{L}(\boldsymbol{w})\lambda$, we have:

$$\frac{1}{T} \sum_{t=0}^{T-1} \min_{\lambda \in \Delta^M} \|\nabla \mathcal{L}(\boldsymbol{w}_t^{(1)})\lambda\|^2 \leq \frac{2\Delta}{\alpha K T} + \frac{2F_1^2}{T} \sum_{t=0}^{T-1} \max_k \left\|\boldsymbol{w}_t^{(k)} - \boldsymbol{w}_t^{(1)}\right\|_\infty^2 + \frac{\alpha^2 F_1(F^2 + \sigma^2)}{2}.$$

That completes our proof. $\qquad\square$

**Lemma A.2.** *In Algorithm 1, if the learning rate $\alpha$ satisfies*

$$\alpha \leq \min\left\{\frac{1}{32nL_\infty}, \frac{1}{16HL_2}\right\},$$

*then the following holds:*

$$\Delta_k \leq 2\alpha H\varsigma + (8\alpha n L_\infty + 4\alpha H L_2)\Delta_{k-1} + 2\alpha n \|\nabla\mathcal{L}(\boldsymbol{w}_k)\|_\infty, \forall k \geq 2,$$

*and*

$$\Delta_1^2 \leq 8\alpha^2 n^2 \|\nabla\mathcal{L}(\boldsymbol{w}_1)\|_\infty^2 + 8\alpha^2 n^2 \varsigma^2.$$

*Finally,*

$$\sum_{k=1}^{K} \Delta_k^2 \leq 16\alpha^2 n^2 \varsigma^2 + 48\alpha^2 H^2 \varsigma^2 K + 48\alpha^2 n^2 \sum_{k=1}^{K} \|\nabla\mathcal{L}(\boldsymbol{w}_k)\|_\infty^2.$$

*Proof.* Without loss of generality, for all $m \in \{2, \cdots, n+1\}$ and all $k \in \{2, \cdots, K\}$,

$$
\begin{aligned}
\boldsymbol{w}_k^{(m)} &= \boldsymbol{w}_k - \alpha \sum_{t=1}^{m-1} \nabla f\left(\boldsymbol{w}_k^{(t)}; \boldsymbol{x}_{\sigma_k(t)}\right) \\
&= \boldsymbol{w}_k - \alpha \sum_{t=1}^{m-1} \nabla f\left(\boldsymbol{w}_{k-1}^{(\sigma_{k-1}^{-1}(\sigma_k(t)))}; \boldsymbol{x}_{\sigma_k(t)}\right) \\
&\quad - \alpha \sum_{t=1}^{m-1} \left(\nabla f\left(\boldsymbol{w}_k^{(t)}; \boldsymbol{x}_{\sigma_k(t)}\right) - \nabla f\left(\boldsymbol{w}_{k-1}^{(\sigma_{k-1}^{-1}(\sigma_k(t)))}; \boldsymbol{x}_{\sigma_k(t)}\right)\right).
\end{aligned}
$$

Now add and subtract

$$\alpha \sum_{t=1}^{m-1} \frac{1}{n} \sum_{s=1}^{n} \nabla f\left(\boldsymbol{w}_{k-1}^{(s)}; \boldsymbol{x}_{\sigma_{k-1}(s)}\right) = \frac{\alpha(m-1)}{n} \sum_{t=1}^{n} \nabla f\left(\boldsymbol{w}_{k-1}^{(t)}; \boldsymbol{x}_{\sigma_{k-1}(t)}\right),$$

which gives

$$
\begin{aligned}
\boldsymbol{w}_k^{(m)} &= \boldsymbol{w}_k - \alpha \sum_{t=1}^{m-1} \left(\nabla f\left(\boldsymbol{w}_{k-1}^{(\sigma_{k-1}^{-1}(\sigma_k(t)))}; \boldsymbol{x}_{\sigma_k(t)}\right) - \frac{1}{n} \sum_{s=1}^{n} \nabla f\left(\boldsymbol{w}_{k-1}^{(s)}; \boldsymbol{x}_{\sigma_{k-1}(s)}\right)\right) \\
&\quad - \frac{\alpha(m-1)}{n} \sum_{t=1}^{n} \nabla f\left(\boldsymbol{w}_{k-1}^{(t)}; \boldsymbol{x}_{\sigma_{k-1}(t)}\right) \\
&\quad - \alpha \sum_{t=1}^{m-1} \left(\nabla f\left(\boldsymbol{w}_k^{(t)}; \boldsymbol{x}_{\sigma_k(t)}\right) - \nabla f\left(\boldsymbol{w}_{k-1}^{(\sigma_{k-1}^{-1}(\sigma_k(t)))}; \boldsymbol{x}_{\sigma_k(t)}\right)\right).
\end{aligned}
$$

We further add and subtract

$$\frac{\alpha(m-1)}{K} \sum_{k=1}^{K} \nabla\mathcal{L}(\boldsymbol{w}_t; \boldsymbol{x}_{\sigma_{t-1}(k)}) = \alpha(m-1)\nabla\mathcal{L}(\boldsymbol{w}_k)$$

to arrive at

$$
\begin{aligned}
\boldsymbol{w}_k^{(m)} \;=\; & \boldsymbol{w}_k - \alpha \sum_{t=1}^{m-1} \left( \nabla f \left( \boldsymbol{w}_{k-1}^{(\sigma_{k-1}^{-1}(\sigma_k(t)))}; \boldsymbol{x}_{\sigma_k(t)} \right) - \frac{1}{n} \sum_{s=1}^{n} \nabla f \left( \boldsymbol{w}_{k-1}^{(s)}; \boldsymbol{x}_{\sigma_{k-1}(s)} \right) \right) \\
& -\alpha(m-1)\nabla \mathcal{L}(\boldsymbol{w}_k) + \frac{\alpha(m-1)}{n} \sum_{t=1}^{n} \left( \nabla f \left( \boldsymbol{w}_k; \boldsymbol{x}_{\sigma_{k-1}(t)} \right) - \nabla f \left( \boldsymbol{w}_{k-1}^{(t)}; \boldsymbol{x}_{\sigma_{k-1}(t)} \right) \right) \\
& -\alpha \sum_{t=1}^{m-1} \left( \nabla f \left( \boldsymbol{w}_k^{(t)}; \boldsymbol{x}_{\sigma_k(t)} \right) - \nabla f \left( \boldsymbol{w}_{k-1}^{(\sigma_{k-1}^{-1}(\sigma_k(t)))}; \boldsymbol{x}_{\sigma_k(t)} \right) \right).
\end{aligned}
$$

We now rearrange, take norms on both sides and apply the triangle inequality,

$$
\begin{aligned}
\left\| \boldsymbol{w}_k^{(m)} - \boldsymbol{w}_k \right\|_\infty \;\leq\; & \alpha \left\| \sum_{t=1}^{m-1} \left( \nabla f \left( \boldsymbol{w}_{k-1}^{(\sigma_{k-1}^{-1}(\sigma_k(t)))}; \boldsymbol{x}_{\sigma_k(t)} \right) - \frac{1}{n} \sum_{s=1}^{n} \nabla f \left( \boldsymbol{w}_{k-1}^{(s)}; \boldsymbol{x}_{\sigma_{k-1}(s)} \right) \right) \right\|_\infty \\
& +\alpha(m-1)\| \nabla \mathcal{L}(\boldsymbol{w}_k) \|_\infty \\
& +\frac{\alpha(m-1)}{n} \left\| \sum_{t=1}^{n} \left( \nabla f \left( \boldsymbol{w}_k; \boldsymbol{x}_{\sigma_{k-1}(t)} \right) - \nabla f \left( \boldsymbol{w}_{k-1}^{(t)}; \boldsymbol{x}_{\sigma_{k-1}(t)} \right) \right) \right\|_\infty \\
& +\alpha \left\| \sum_{t=1}^{m-1} \left( \nabla f \left( \boldsymbol{w}_k^{(t)}; \boldsymbol{x}_{\sigma_k(t)} \right) - \nabla f \left( \boldsymbol{w}_{k-1}^{(\sigma_{k-1}^{-1}(\sigma_k(t)))}; \boldsymbol{x}_{\sigma_k(t)} \right) \right) \right\|_\infty . \quad (9)
\end{aligned}
$$

There are four terms on the right hand side. We apply Assumption 3.7 on the first term, and Assumption 3.4 on the last two terms. First, for the first term,

$$
\begin{aligned}
& \left\| \nabla f \left( \boldsymbol{w}_{k-1}^{(\sigma_{k-1}^{-1}(\sigma_k(t)))}; \boldsymbol{x}_{\sigma_k(t)} \right) - \frac{1}{n} \sum_{s=1}^{n} \nabla f \left( \boldsymbol{w}_{k-1}^{(s)}; \boldsymbol{x}_{\sigma_{k-1}(s)} \right) \right\| \\
\leq\; & \left\| \nabla f \left( \boldsymbol{w}_{k-1}^{(\sigma_{k-1}^{-1}(\sigma_k(t)))}; \boldsymbol{x}_{\sigma_k(t)} \right) - \frac{1}{n} \sum_{s=1}^{n} \nabla f \left( \boldsymbol{w}_{k-1}^{(\sigma_{k-1}^{-1}(\sigma_k(t)))}; \boldsymbol{x}_{\sigma_{k-1}(s)} \right) \right\| \\
& +\left\| \frac{1}{n} \sum_{s=1}^{n} \nabla f \left( \boldsymbol{w}_{k-1}^{(\sigma_{k-1}^{-1}(\sigma_k(t)))}; \boldsymbol{x}_{\sigma_{k-1}(s)} \right) - \frac{1}{n} \sum_{s=1}^{n} \nabla f \left( \boldsymbol{w}_{k-1}^{(s)}; \boldsymbol{x}_{\sigma_{k-1}(s)} \right) \right\| \\
\overset{\text{Assumption 3.4 and 3.5}}{\leq}\; & \varsigma + \sigma_i + \frac{L_2}{n} \sum_{s=1}^{n} \left\| \boldsymbol{w}_{k-1}^{(\sigma_{k-1}^{-1}(\sigma_k(t)))} - \boldsymbol{w}_{k-1}^{(s)} \right\|_\infty \\
\leq\; & \max_m \sigma_m + \frac{L_2}{n} \sum_{s=1}^{n} \left( \left\| \boldsymbol{w}_{k-1} - \boldsymbol{w}_{k-1}^{(\sigma_{k-1}^{-1}(\sigma_k(t)))} \right\|_\infty + \left\| \boldsymbol{w}_{k-1} - \boldsymbol{w}_{k-1}^{(s)} \right\|_\infty \right) \\
\leq\; & \max_m \sigma_m + 2L_2 \Delta_{k-1}.
\end{aligned}
$$

Denote

$$
\boldsymbol{u}_t := \nabla \ell \left( \boldsymbol{w}_{k-1}^{\sigma_{k-1}^{-1}(\sigma_k(t))}; \boldsymbol{x}_{\sigma_k(t)} \right) - \frac{1}{n} \sum_{s=1}^{n} \nabla \mathcal{L}(\boldsymbol{w}_{k-1}^{(s)}; \boldsymbol{x}_{\sigma_{k-1}(s)}).
$$

We can use Assumption 3.7 to obtain a bound on the prefix sum

$$
\left\| \sum_{t=1}^{m-1} \frac{\boldsymbol{u}_t}{\varsigma + \sigma_i + 2L_2 \Delta_{k-1}} \right\|_\infty \leq A,
$$

that is,

$$
\left\| \sum_{t=1}^{m-1} \left( \nabla f \left( \boldsymbol{w}_{k-1}^{(\sigma_{k-1}^{-1}(\sigma_k(t)))}; \boldsymbol{x}_{\sigma_k(t)} \right) - \frac{1}{n} \sum_{s=1}^{n} \nabla f \left( \boldsymbol{w}_{k-1}^{(s)}; \boldsymbol{x}_{\sigma_{k-1}(s)} \right) \right) \right\|_\infty \leq A(\varsigma + \sigma_i + 2L_2 \Delta_{k-1}).
$$

Now we have a bound for the first term in Equation (9), we proceed to bound the last two terms where we apply Assumption 3.4. We can then rewrite Equation (9) into

$$
\begin{aligned}
\left\| \boldsymbol{w}_k^{(m)} - \boldsymbol{w}_k \right\|_\infty \leq\ & \alpha A(\varsigma + \sigma_i + 2L_2\Delta_{k-1}) + \alpha(m-1)\|\nabla\mathcal{L}(\boldsymbol{w}_k)\|_\infty \\
& + \frac{\alpha L_\infty(m-1)}{n}\sum_{t=1}^{n}\left\|\boldsymbol{w}_k - \boldsymbol{w}_{k-1}^{(t)}\right\|_\infty + \alpha L_\infty\sum_{t=1}^{m-1}\left\|\boldsymbol{w}_k^{(t)} - \boldsymbol{w}_{k-1}^{(\sigma_{k-1}^{-1}(\sigma_k(t)))}\right\|_\infty .
\end{aligned}
$$

Furthermore, applying the triangle inequality to the norms in the last two terms, we obtain

$$
\left\|\boldsymbol{w}_{k-1}^{(t)} - \boldsymbol{w}_k\right\|_\infty = \left\|\boldsymbol{w}_{k-1}^{(t)} - \boldsymbol{w}_{k-1} + \boldsymbol{w}_{k-1} - \boldsymbol{w}_{k-1}^{(n+1)}\right\|_\infty \leq 2\Delta_{k-1},
$$

and similarly,

$$
\begin{aligned}
\left\|\boldsymbol{w}_k^{(t)} - \boldsymbol{w}_{k-1}^{(\sigma_{k-1}^{-1}(\sigma_k(t)))}\right\|_\infty &= \left\|\boldsymbol{w}_k^{(t)} - \boldsymbol{w}_k + \boldsymbol{w}_k - \boldsymbol{w}_{k-1} + \boldsymbol{w}_{k-1} - \boldsymbol{w}_{k-1}^{(\sigma_{k-1}^{-1}(\sigma_k(t)))}\right\|_\infty \\
&\leq \Delta_k + 2\Delta_{k-1}.
\end{aligned}
$$

This gives

$$
\begin{aligned}
\left\|\boldsymbol{w}_k^{(m)} - \boldsymbol{w}_k\right\|_\infty \leq\ & \alpha A(\varsigma + \sigma_i + 2L_2\Delta_{k-1}) + \alpha(m-1)\|\nabla\mathcal{L}(\boldsymbol{w}_k)\|_\infty + 2\alpha L_\infty(m-1)\Delta_{k-1} \\
& + \alpha L_\infty(m-1)(2\Delta_{k-1} + \Delta_k) \\
\leq\ & \alpha A(\varsigma + \sigma_i + 2L_2\Delta_{k-1}) + \alpha(m-1)\|\nabla\mathcal{L}(\boldsymbol{w}_k)\|_\infty \\
& + \alpha L_\infty(m-1)(4\Delta_{k-1} + \Delta_k). \qquad (10)
\end{aligned}
$$

Note that Equation (10) only holds with $k \in \{2, \ldots, K\}$ and $m \in \{2, \ldots, n+1\}$. We now discuss the boundary cases. Note that the bound of Equation (10) trivially holds with $m = 1$ for any $k$ since the left hand side becomes zero. On the other hand, when $k = 1$, we have

$$
\begin{aligned}
\boldsymbol{w}_1^{(m)} =\ & \boldsymbol{w}_1 - \alpha\sum_{t=1}^{m-1}\nabla f\left(\boldsymbol{w}_1^{(t)};\boldsymbol{x}_{\sigma_1(t)}\right) \\
=\ & \boldsymbol{w}_1 - \alpha\sum_{t=1}^{m-1}\frac{1}{n}\sum_{s=1}^{n}\nabla f\left(\boldsymbol{w}_1;\boldsymbol{x}_{\sigma_1(s)}\right) + \alpha\sum_{t=1}^{m-1}\nabla f\left(\boldsymbol{w}_1^{(t)};\boldsymbol{x}_{\sigma_1(t)}\right) \\
& -\alpha\sum_{t=1}^{m-1}\nabla f\left(\boldsymbol{w}_1;\boldsymbol{x}_{\sigma_1(t)}\right) + \alpha\sum_{t=1}^{m-1}\nabla f\left(\boldsymbol{w}_1;\boldsymbol{x}_{\sigma_1(t)}\right) - \alpha\sum_{t=1}^{m-1}\frac{1}{n}\sum_{s=1}^{n}\nabla f\left(\boldsymbol{w}_1;\boldsymbol{x}_{\sigma_1(s)}\right).
\end{aligned}
$$

Take norms and apply the triangle inequality, we obtain

$$
\begin{aligned}
\left\|\boldsymbol{w}_1^{(m)} - \boldsymbol{w}_1\right\|_\infty \leq\ & \alpha\left\|\sum_{t=1}^{m-1}\frac{1}{n}\sum_{s=1}^{n}\nabla f\left(\boldsymbol{w}_1;\boldsymbol{x}_{\sigma_1(s)}\right)\right\|_\infty \\
& + \alpha\left\|\sum_{t=1}^{m-1}\left(\nabla f\left(\boldsymbol{w}_1^{(t)};\boldsymbol{x}_{\sigma_1(t)}\right) - \nabla f\left(\boldsymbol{w}_1;\boldsymbol{x}_{\sigma_1(s)}\right)\right)\right\|_\infty \\
& + \alpha\left\|\sum_{t=1}^{m-1}\left(\nabla f\left(\boldsymbol{w}_1;\boldsymbol{x}_{\sigma_1(t)}\right) - \frac{1}{n}\sum_{s=1}^{n}\nabla f\left(\boldsymbol{w}_1;\boldsymbol{x}_{\sigma_1(s)}\right)\right)\right\|_\infty \\
\leq\ & \alpha(m-1)\|\nabla\mathcal{L}(\boldsymbol{w}_1)\|_\infty + \alpha(m-1)L_\infty\Delta_1 + \alpha(m-1)(\varsigma + \sigma_i) \\
\leq\ & \alpha n\|\nabla\mathcal{L}(\boldsymbol{w}_1)\|_\infty + \alpha n L_\infty\Delta_1 + \alpha n(\varsigma + \sigma_i). \qquad (11)
\end{aligned}
$$

Now that we have the bounds for $\Delta_k$, we next will sum them up. Taking a max over $m$ on both sides in Equation (10), this implies for all the $k \geq 2$,

$$
\Delta_k \leq \alpha H(\varsigma + \sigma_i + 2L_2\Delta_{k-1}) + \alpha L_\infty n(4\Delta_{k-1} + \Delta_k) + \alpha n\|\nabla\mathcal{L}(\boldsymbol{w}_k)\|_\infty,
$$

as $m - 1 \leq n$. Considering the fact that $\alpha L_\infty n < 1/2$, we get

$$
\Delta_k \leq 2\alpha H\varsigma + \sigma_i + (8\alpha n L_\infty + 4\alpha H L_2)\Delta_{k-1} + 2\alpha n\|\nabla\mathcal{L}(\boldsymbol{w}_k)\|_\infty.
$$

This completes the proof of the first inequality in the lemma. Applying this recursively from any $k \geq 2$ to 2, this gives

$$\Delta_k \leq (8\alpha n L_\infty + 4\alpha H L_2)^{k-1} \Delta_1 + \sum_{i=1}^\infty (8\alpha n L_\infty + 4\alpha H L_2)^i \left(2\alpha H(\varsigma + \sigma_i) + 2\alpha n \|\nabla\mathcal{L}(\boldsymbol{w}_k)\|_\infty\right).$$

Applying the learning rate conditions that $32\alpha n L_\infty \leq 1$ and $16\alpha H L_2 \leq 1$, we obtain

$$\Delta_k \leq \left(\frac{1}{2}\right)^{k-1} \Delta_1 + 4\alpha H(\varsigma + \sigma_i) + 4\alpha n \|\nabla\mathcal{L}(\boldsymbol{w}_k)\|_\infty.$$

Square on both sides,

$$\Delta_k^2 \leq 3\left(\frac{1}{4}\right)^{k-1} \Delta_1^2 + 48\alpha^2 H^2(\varsigma + \sigma_i)^2 + 48\alpha^2 n^2 \|\nabla\mathcal{L}(\boldsymbol{w}_k)\|_\infty^2.$$

We can apply a similar trick to Equation (11) and get

$$\Delta_1^2 \leq 8\alpha^2 n^2 \|\nabla\mathcal{L}(\boldsymbol{w}_1)\|_\infty^2 + 8\alpha^2 n^2(\varsigma + \sigma_i)^2.$$

This completes the proof of the second inequality in the lemma. Summing from $k = 1$ to $K$, we get

$$
\begin{aligned}
\sum_{k=1}^K \Delta_k^2 &= \Delta_1^2 + \sum_{k=2}^K \Delta_k^2 \\
&= \Delta_1^2 + 3\Delta_1^2 \sum_{k=2}^K \left(\frac{1}{4}\right)^{k-1} + 48\alpha^2 H^2(\varsigma + \sigma_i)^2(K-1) + 48\alpha^2 n^2 \sum_{k=2}^K \|\nabla\mathcal{L}(\boldsymbol{w}_k)\|_\infty^2 \\
&\leq \Delta_1^2 + 3\Delta_1^2 \sum_{k=1}^\infty \left(\frac{1}{4}\right)^k + 48\alpha^2 H^2(\varsigma + \sigma_i)^2(K-1) + 48\alpha^2 n^2 \sum_{k=2}^K \|\nabla\mathcal{L}(\boldsymbol{w}_k)\|_\infty^2 \\
&\leq 16\alpha^2 n^2 \|\nabla\mathcal{L}(\boldsymbol{w}_1)\|_\infty^2 + 16\alpha^2 n^2(\varsigma + \sigma_i)^2 \\
&\quad + 48\alpha^2 H^2(\varsigma + \sigma_i)^2(K-1) + 48\alpha^2 n^2 \sum_{k=2}^K \|\nabla\mathcal{L}(\boldsymbol{w}_k)\|_\infty^2 \\
&\leq 16\alpha^2 n^2(\varsigma + \sigma_i)^2 + 48\alpha^2 H^2(\varsigma + \sigma_i)^2 K + 48\alpha^2 n^2 \sum_{k=1}^K \|\nabla\mathcal{L}(\boldsymbol{w}_k)\|_\infty^2.
\end{aligned}
$$

That completes the third inequality, and we have finished proving all three inequalities. $\square$

**Lemma A.3** ((Chen et al., 2024)). *Given $Q \in \mathbb{R}^{d \times M}$, recall $\lambda_{Q,\rho}^*$ with $\rho \geq 0$ is defined as*

$$\lambda_{Q,\rho}^* \in \arg\min_{\lambda \in \Delta^M} \|Q\lambda\|^2 + \rho\|\lambda\|^2. \tag{12}$$

*Then, for any $\lambda \in \Delta^M$, it holds that*

$$\langle Q\lambda_{Q,\rho}^*, Q\lambda \rangle \geq \|Q\lambda_{Q,\rho}^*\|^2 - \rho, \tag{13}$$

$$\text{and } \|Q\lambda - Q\lambda_{Q,\rho}^*\|^2 \leq \|Q\lambda\|^2 - \|Q\lambda_{Q,\rho}^*\|^2 + 2\rho. \tag{14}$$

*Proof.* By the first-order optimality condition for equation (12), for any $\lambda \in \Delta^M$, we have

$$\langle Q^\top Q\lambda_{Q,\rho}^*, \lambda - \lambda_{Q,\rho}^* \rangle \geq -\rho.$$

By rearranging the above inequality, we obtain

$$\langle Q\lambda_{Q,\rho}^*, Q\lambda \rangle \geq \|Q\lambda_{Q,\rho}^*\|^2 - \rho,$$

which is precisely the first inequality in the claim. Furthermore, we also have

$$
\begin{aligned}
\|Q\lambda - Q\lambda_{Q,\rho}^*\|^2 &= \|Q\lambda\|^2 + \|Q\lambda_{Q,\rho}^*\|^2 - 2\langle Q\lambda_{Q,\rho}^*, Q\lambda \rangle \\
&\leq \|Q\lambda\|^2 + \|Q\lambda_{Q,\rho}^*\|^2 - 2\|Q\lambda_{Q,\rho}^*\|^2 + 2\rho \\
&= \|Q\lambda\|^2 - \|Q\lambda_{Q,\rho}^*\|^2 + 2\rho,
\end{aligned}
$$

which is the desired second inequality in the claim. Hence, the proof is complete. $\square$

**Lemma A.4** (Hölder continuity of $d_Q$ w.r.t. $Q$ (Chen et al., 2024))**.** *For all $Q, Q' \in \mathbb{R}^{d \times M}$, define $\lambda^* \in \arg\min_{\lambda \in \Delta^M} \|Q\lambda\|^2$, and $\lambda^{*\prime} \in \arg\min_{\lambda \in \Delta^M} \|Q'\lambda\|^2$, and $d_Q = Q\lambda^*$, $d_Q' = Q'\lambda^{*\prime}$, then*

$$\|d_Q - d_{Q'}\|^2 \le 4 \max \left\{ \sup_{\lambda \in \Delta^M} \|Q\lambda\|, \sup_{\lambda \in \Delta^M} \|Q'\lambda\| \right\} \cdot \sup_{\lambda \in \Delta^M} \|(Q - Q')\lambda\|. \tag{15}$$

*Proof.* We first rewrite $\|d_Q - d_{Q'}\|^2 = \|Q\lambda^* - Q'\lambda^{*\prime}\|^2$ as

$$
\begin{aligned}
\|Q\lambda^* - Q'\lambda^{*\prime}\|^2 =& \|Q\lambda^*\|^2 + \|Q'\lambda^{*\prime}\|^2 - 2\langle Q\lambda^*, Q'\lambda^{*\prime} \rangle \\
=& \|Q\lambda^*\|^2 - \|Q'\lambda^{*\prime}\|^2 + 2\langle Q'\lambda^{*\prime}, Q'\lambda^{*\prime} - Q\lambda^* \rangle \\
=& \|Q\lambda^*\|^2 - \|Q'\lambda^{*\prime}\|^2 + \underbrace{2\langle Q'\lambda^{*\prime}, Q'\lambda^{*\prime} - Q'\lambda^* \rangle}_{\le 0} + 2\langle Q'\lambda^{*\prime}, Q'\lambda^* - Q\lambda^* \rangle,
\end{aligned}
$$

where $\langle Q'\lambda^{*\prime}, Q'\lambda^{*\prime} - Q'\lambda^* \rangle \le 0$ by (13) in Lemma A.3. Then it can be further bounded by

$$
\begin{aligned}
\|Q\lambda^* - Q'\lambda^{*\prime}\|^2 &\overset{(a)}{\le} \min_{\lambda \in \Delta^M} \|Q\lambda\|^2 - \min_{\lambda \in \Delta^M} \|Q'\lambda\|^2 + 2\|Q'\lambda^{*\prime}\|\|(Q' - Q)\lambda^*\| \\
&= -\max_{\lambda \in \Delta^M} -\|Q\lambda\|^2 + \max_{\lambda \in \Delta^M} -\|Q'\lambda\|^2 + 2\|Q'\lambda^{*\prime}\|\|(Q' - Q)\lambda^*\| \\
&\overset{(b)}{\le} \max_{\lambda \in \Delta^M} \left( \|Q\lambda\|^2 - \|Q'\lambda\|^2 \right) + 2\|Q'\lambda^{*\prime}\|\|(Q' - Q)\lambda^*\| \\
&\overset{(c)}{\le} \max_{\lambda \in \Delta^M} \|(Q - Q')\lambda\| \left( \|Q\lambda\| + \|Q'\lambda\| \right) + 2\|Q'\lambda^{*\prime}\|\|(Q' - Q)\lambda^*\| \\
&\le 4 \max \left\{ \sup_{\lambda \in \Delta^M} \|Q\lambda\|, \sup_{\lambda \in \Delta^M} \|Q'\lambda\| \right\} \cdot \sup_{\lambda \in \Delta^M} \|(Q - Q')\lambda\|,
\end{aligned}
$$

where $(a)$ follows from the Cauchy-Schwartz inequality; $(b)$ follows from subadditivity of maximum operator; $(c)$ follows from the triangle inequality. The proof is complete. $\square$

## A.6  PROOF ON THE CONVERGENCE RATE OF ALGORITHM 1 WITH RANDOM ORDERING

The following Theorem studies the convergence rate of Algorithm 1 with random ordering.

**Theorem A.5.** *Set $\alpha = \min \left\{ \sqrt{\frac{24\Delta}{KLT \sum_{k=1}^{K} \sigma_k^2}}, \frac{1}{\sqrt{2}KL}, \frac{1}{AL^2 K^2 T^{1/3}} \right\}$ with* random *yields:*

$$\frac{1}{T} \sum_{t=0}^{T-1} \mathbb{E}\|\nabla \mathcal{L}(\boldsymbol{w}_t)\|_2^2 \le \sqrt{\frac{24L\Delta}{KT} \sum_{k=1}^{K} \sigma_k^2} + \frac{48L\Delta B^2}{\frac{T}{K} \sum_{k=1}^{K} \sigma_k^2}.$$

To prove Theorem A.5, we first need the following Lemma:

**Lemma A.6.** *Suppose that Assumption 3.4 holds. Then for iterates $\boldsymbol{w}_t$ generated by Algorithm 1 with stepsize $\alpha \le \frac{1}{Ln}$, we have*

$$\mathcal{L}(\boldsymbol{w}_{t+1}) \le \mathcal{L}(\boldsymbol{w}_t) - \frac{\alpha K}{2} \|\nabla \mathcal{L}(\boldsymbol{w}_t)\|^2 + \frac{\alpha L_2^2}{K} V_i + \frac{\alpha^2 L}{2} \sum_{k=1}^{K} \sigma_k^2, \tag{16}$$

*where $V_t \equiv \sum_{k=1}^{K} \left\| \boldsymbol{w}_t - \boldsymbol{w}_t^{(k)} \right\|_\infty^2$.*

*Proof.* Recall that $\boldsymbol{w}_{t+1} = \boldsymbol{w}_t - \alpha g_t$, where $g_t = \sum_{i=0}^{n-1} \nabla f_{\pi_i}(\boldsymbol{w}_t^i)$. Using $L$-smoothness of $f$, we get

$$
\begin{aligned}
\mathbb{E}\mathcal{L}(\boldsymbol{w}_{t+1}) \;\leq\;& \mathbb{E}\mathcal{L}(\boldsymbol{w}_t) - \alpha K \mathbb{E}\left\langle \nabla \mathcal{L}(\boldsymbol{w}_k), \frac{1}{K}\sum_{k=1}^{K} \nabla \ell(\boldsymbol{w}_k^{(t)}; \xi_{\sigma_k(t)}) \right\rangle \\
&+ \frac{\alpha^2 K^2 L}{2}\mathbb{E}\left\| \frac{1}{n}\sum_{t=1}^{n} \nabla \ell(\boldsymbol{w}_k^{(t)}; \xi_{\sigma_k(t)}) \right\|^2 \\
=\;& \mathbb{E}\mathcal{L}(\boldsymbol{w}_k) - \frac{\alpha K}{2}\|\nabla\mathcal{L}(\boldsymbol{w}_k)\|^2 - \frac{\alpha n}{2}\left\| \frac{1}{K}\sum_{k=1}^{K} \nabla\mathcal{L}_k(\boldsymbol{w}_k^{(t)}) \right\|^2 \\
&+ \frac{\alpha K}{2}\left\| \nabla\mathcal{L}(\boldsymbol{w}_k) - \frac{1}{K}\sum_{k=1}^{K} \nabla\mathcal{L}_k(\boldsymbol{w}_k^{(t)}) \right\|^2 + \frac{\alpha^2 n^2 L}{2}\mathbb{E}\left\| \frac{1}{K}\sum_{k=1}^{K}\nabla\ell(\boldsymbol{w}_k^{(t)};\xi_{\sigma_k(t)}) \right\|^2 \\
\leq\;& \mathbb{E}\mathcal{L}(\boldsymbol{w}_k) - \frac{\alpha K}{2}\|\nabla\mathcal{L}(\boldsymbol{w}_k)\|^2 + \frac{\alpha n}{2}\left\| \nabla\mathcal{L}(\boldsymbol{w}_k) - \frac{1}{K}\sum_{k=1}^{K}\nabla\mathcal{L}_k(\boldsymbol{w}_k^{(t)}) \right\|^2 + \frac{\alpha^2 L}{2}\sum_{k=1}^{K}\sigma_k^2.
\end{aligned}
$$

Then note that

$$
\begin{aligned}
\left\| \nabla\mathcal{L}(\boldsymbol{w}_t) - \frac{1}{K}\sum_{k=1}^{K}\nabla\mathcal{L}_k(\boldsymbol{w}_t^{(k)}) \right\|^2 \;=\;& \left\| \frac{1}{K}\sum_{k=1}^{K}\nabla\mathcal{L}_k(\boldsymbol{w}_t) - \frac{1}{K}\sum_{k=1}^{K}\nabla\mathcal{L}_k(\boldsymbol{w}_t^{(k)}) \right\|^2 \\
\leq\;& \frac{1}{K}\sum_{k=1}^{K}\left\| \nabla\mathcal{L}_k(\boldsymbol{w}_t) - \nabla\mathcal{L}_k(\boldsymbol{w}_t^{(k)}) \right\|^2 \\
\leq\;& \frac{1}{K}\sum_{k=1}^{K}L_2^2\left\| \boldsymbol{w}_t - \boldsymbol{w}_t^{(k)} \right\|_\infty^2 \\
\leq\;& \frac{L_2^2}{K}V_i,
\end{aligned}
$$

which completes the proof. $\qquad\square$

**Lemma A.7.** *Suppose that Assumption 3.4 holds and that Algorithm 1 is used with a stepsize $\alpha \leq \frac{1}{2LK}$. Then*

$$
\mathbb{E}[V_t] \leq \alpha^2 K^3 \|\nabla f(\boldsymbol{w}_t)\|^2 + \alpha^2 K^2 \varsigma^2, \tag{17}
$$

*where $V_t \equiv \sum_{k=1}^{K}\|\boldsymbol{w}_t - \boldsymbol{w}_t^{(k)}\|_\infty^2$.*

*Proof.* Let us fix any $k \in [1, K-1]$ and find an upper bound for $\mathbb{E}_t\|\boldsymbol{w}_t^k - \boldsymbol{w}_t\|^2$. First, note that

$$
\boldsymbol{w}_t^k = \boldsymbol{w}_t - \alpha\sum_{i=0}^{k-1}\nabla\ell(\boldsymbol{w}_t^i, \xi_t^i).
$$

Therefore, by Young's inequality, Jensen's inequality and gradient Lipschitzness,

$$
\begin{aligned}
\mathbb{E}_t\|\boldsymbol{w}_t^k - \boldsymbol{w}_t\|^2 \;=\;& \alpha^2 \mathbb{E}_t\left\| \sum_{i=0}^{k-1}\nabla\ell(\boldsymbol{w}_t^i, \xi_t^i) \right\|^2 \\
\leq\;& 2\alpha^2 \mathbb{E}_t\left\| \sum_{i=0}^{k-1}\left(\nabla\ell(\boldsymbol{w}_t^i, \xi_t^i) - \nabla\ell(\boldsymbol{w}_t, \xi_t^i)\right) \right\|^2 + 2\alpha^2 \mathbb{E}_t\left\| \sum_{i=0}^{k-1}\nabla\ell(\boldsymbol{w}_t, \xi_t^i) \right\|^2 \\
\leq\;& 2\alpha^2 k \sum_{i=0}^{k-1}\mathbb{E}_t\|\nabla\ell(\boldsymbol{w}_t^i, \xi_t^i) - \nabla\ell(\boldsymbol{w}_t, \xi_t^i)\|^2 + 2\alpha^2 \mathbb{E}_t\left\| \sum_{i=0}^{k-1}\nabla\ell(\boldsymbol{w}_t, \xi_t^i) \right\|^2 \\
\leq\;& 2\alpha^2 L^2 k \sum_{i=0}^{k-1}\mathbb{E}_t\|\boldsymbol{w}_t^i - \boldsymbol{w}_t\|^2 + 2\alpha^2 \mathbb{E}_t\left\| \sum_{i=0}^{k-1}\nabla\ell(\boldsymbol{w}_t, \xi_t^i) \right\|^2.
\end{aligned}
$$

Let us bound the second term. For any $i$ we have $\mathbb{E}_t[\nabla\ell(\boldsymbol{w}_t, \xi_t^i)] = \nabla\mathcal{L}(\boldsymbol{w}_t)$. So using (with vectors $\nabla f_{\pi_0}(x_t), \nabla f_{\pi_1}(x_t), \ldots, \nabla f_{\pi_{k-1}}(x_t)$), we obtain

$$\mathbb{E}_t \left\| \sum_{i=0}^{k-1} \nabla\ell(\boldsymbol{w}_t, \xi_t^i) \right\|^2 = k^2 \|\nabla\mathcal{L}(\boldsymbol{w}_t)\|^2 + k^2 \mathbb{E}_t \left\| \frac{1}{k} \sum_{i=0}^{k-1} (\nabla\ell(\boldsymbol{w}_t, \xi_t^i) - \nabla\mathcal{L}(\boldsymbol{w}_t)) \right\|^2$$

$$\leq k^2 \|\nabla\mathcal{L}(\boldsymbol{w}_t)\|^2 + \frac{k(K-k)}{K-1}(\varsigma + \max_k \sigma_k)^2.$$

Combining the produced bounds yields

$$\mathbb{E}_t \|\boldsymbol{w}_t^k - \boldsymbol{w}_t\|^2 \leq 2\alpha^2 L^2 k \sum_{i=0}^{k-1} \mathbb{E}_t \|\boldsymbol{w}_t^i - \boldsymbol{w}_t\|^2 + 2\alpha^2 k^2 \|\nabla f(x_t)\|^2 + 2\alpha^2 \frac{k(K-k)}{K-1}(\varsigma + \max_k \sigma_k)^2$$

$$\leq 2\alpha^2 L^2 k \mathbb{E}[V_t] + 2\alpha^2 k^2 \|\nabla f(x_t)\|^2 + 2\alpha^2 \frac{k(K-k)}{K-1}(\varsigma + \max_k \sigma_k)^2,$$

whence

$$\mathbb{E}[V_t] = \sum_{k=0}^{K-1} \mathbb{E}_t \|\boldsymbol{w}_t^k - \boldsymbol{w}_t\|^2$$

$$\leq \alpha^2 L^2 K(K-1)\mathbb{E}[V_t] + \frac{1}{3}\alpha^2 (K-1)K(2K-1)\|\nabla f(x_t)\|^2 + \frac{1}{3}\alpha^2 K(K+1)(\varsigma + \max_k \sigma_k)^2.$$

Since $\mathbb{E}[V_t]$ appears in both sides of the equation, we rearrange and use that $\alpha \leq \frac{1}{2LK}$ by assumption, which leads to

$$\mathbb{E}[V_t] \leq \frac{4}{3}(1 - \alpha^2 L^2 n(n-1))\mathbb{E}[V_t]$$

$$\leq \frac{4}{9}\alpha^2(n-1)n(2n-1)\|\nabla\mathcal{L}(\boldsymbol{w}_t)\|^2 + \frac{4}{9}\alpha^2 n(n+1)\sigma_t^2$$

$$\leq \alpha^2 n^3 \|\nabla\mathcal{L}(\boldsymbol{w}_t)\|^2 + \alpha^2 n^2 (\varsigma + \max_k \sigma_k)^2.$$

$\square$

Now we are ready to prove theorem A.5:

*Proof.* Taking expectation in Lemma A.6 and then using A.7, we have that for any $t \in \{0, 1, \ldots, T-1\}$,

$$\mathbb{E}_t[\mathcal{L}(\boldsymbol{w}_{t+1})] \overset{(16)}{\leq} \mathcal{L}(\boldsymbol{w}_t) - \frac{\alpha K}{2}\|\nabla\mathcal{L}(\boldsymbol{w}_t)\|^2 + \alpha L^2 \mathbb{E}_t[V_t] + \frac{\alpha^2 L}{2}\sum_{k=1}^{K}\sigma_k^2$$

$$\overset{(17)}{\leq} \mathcal{L}(\boldsymbol{w}_t) - \frac{\alpha K}{2}\|\nabla\mathcal{L}(\boldsymbol{w}_t)\|^2 + \alpha L^2(\alpha^2 K^3 \|\nabla\mathcal{L}(\boldsymbol{w}_t)\|^2 + \alpha^2 K^2(\varsigma + \max_k \sigma_k)^2)$$

$$+ \frac{\alpha^2 L}{2}\sum_{k=1}^{K}\sigma_k^2$$

$$= \mathcal{L}(\boldsymbol{w}_t) - \frac{\alpha K}{2}(1 - \alpha^2 L^2 K^2)\|\nabla\mathcal{L}(\boldsymbol{w}_t)\|^2 + \alpha^3 L^2 K^2(\varsigma + \max_k \sigma_k)^2 + \frac{\alpha^2 L}{2}\sum_{k=1}^{K}\sigma_k^2.$$

Let $\delta_t = \mathcal{L}(\boldsymbol{w}_t) - \mathcal{L}^*$. Adding $-\mathcal{L}^*$ to both sides gives:

$$\mathbb{E}_t[\delta_{t+1}] \leq \delta_t - \frac{\alpha K}{2}(1 - \alpha^2 L^2 K^2)\|\nabla\mathcal{L}(\boldsymbol{w}_t)\|^2 + \alpha^3 L^2 K^2(\varsigma + \max_k \sigma_k)^2 + \frac{\alpha^2 L}{2}\sum_{k=1}^{K}\sigma_k^2$$

$$\leq (1 + \alpha^3 A L^2 K^2)\delta_t - \frac{\alpha K}{2}(1 - \alpha^2 L^2 K^2)\|\nabla\mathcal{L}(\boldsymbol{w}_t)\|^2 + \alpha^3 L^2 K^2(\varsigma + \max_k \sigma_k)^2$$

$$+ \frac{\alpha^2 L}{2}\sum_{k=1}^{K}\sigma_k^2.$$

Taking unconditional expectations in the last inequality and using that by assumption on $\alpha$ we have $1 - \alpha^2 L^2 K^2 \geq \frac{1}{2}$, we get the estimate

$$\mathbb{E}[\delta_{t+1}] \leq (1 + \alpha^3 A L^2 K^2)\mathbb{E}(\delta_t) - \frac{\alpha K}{4}\mathbb{E}[\|\nabla\mathcal{L}(\boldsymbol{w}_t)\|^2] + \alpha^3 L^2 K^2 (\varsigma + \max_k \sigma_k)^2 + \frac{\alpha^2 L}{2}\sum_{k=1}^{K}\sigma_k^2.$$

Then we have:

$$\min_{t=0,\ldots,T-1}\mathbb{E}[\|\nabla\mathcal{L}(\boldsymbol{w}_t)\|^2] \quad \leq \quad \frac{4(1 + \alpha^3 A L^2 K^2)^T}{\alpha K T}(\mathcal{L}(\boldsymbol{w}_0) - \mathcal{L}^*)$$

$$+ 2\alpha^2 L^2 K(\varsigma + \max_k \sigma_k)^2 + \frac{\alpha L}{2}\sum_{k=1}^{K}\sigma_k^2.$$

Using that $1 + x \leq \exp(x)$ and that the stepsize $\alpha$ satisfies $\alpha \leq (AL^2 K^2 T)^{-1/3}$, we have

$$(1 + \alpha^3 A L^2 K^2)^T \leq \exp(\alpha^3 A L^2 K^2 T) \leq \exp(1) \leq 3.$$

Using this in the previous bound, we finally obtain

$$\min_{t=0,\ldots,T-1}\mathbb{E}[\|\nabla\mathcal{L}(\boldsymbol{w}_t)\|^2] \leq \frac{12(\mathcal{L}(\boldsymbol{w}_0) - \mathcal{L}^*)}{\alpha K T} + 2\alpha^2 L^2 K(\varsigma + \max_k \sigma_k)^2 + \frac{\alpha L}{2}\sum_{k=1}^{K}\sigma_k^2.$$

$\square$

## B  DETAILS ON EXPERIMENTAL SETUP

All experiments are conducted on a server with an Intel Xeon Gold 6342 CPU and an NVIDIA RTX A6000 GPU. We use the PyTorch version 1.10.1 with CUDA version 11.7. For experiments on the NYUv2 data set, we train a Multi-Task Attention Network (MTAN) (Liu et al., 2019) following previous works on multi-task learning (Yu et al., 2020; Navon et al., 2022). We also follow the training procedure from (Liu et al., 2019; Yu et al., 2020; Navon et al., 2022). Each method is trained for 200 epochs with the Adam optimizer (Kingma & Ba, 2015). We set the learning rate $\alpha = 1 \times 10^{-4}$ at the beginning of training, and reduce it to $5 \times 10^{-5}$ after 100 epochs. The batch size is set to 2 for all methods.

For experiments on QM9 data set, we use the MPNN model proposed in (Gilmer et al., 2017). Each method is trained for 300 epochs with the Adam optimizer (Kingma & Ba, 2015) and we set the learning rate $\alpha = 1 \times 10^{-4}$ throughout the whole training process. The batch size is set to 120 for all methods.

