# OpenReview forum: "Joint Gradient Balancing for Data Ordering in Finite-Sum Multi-Objective Optimization"
_ICLR.cc/2025/Conference — ICLR 2025 Spotlight_

### Official Review · Reviewer_ntx7 · 2024-10-25

**Soundness:** 3
**Presentation:** 2
**Contribution:** 3
**Rating:** 6
**Confidence:** 3

**Summary:**

The authors in this paper modify the classical method in finite-sum multi-objective optimization such as MGDA algorithm by developing a sample ordering rule. JoGBa jointly balances gradients from multiple objectives during optimization. It is demonstrated for both theoretical analysis and empirical results, the proposed sample ordering outperforms random ordering.

**Strengths:**

The main idea of this paper is clearly presented through the incorporation of the MGDA algorithm with a novel sample ordering method. Figure 1 offers a great visualization and improves the audience's understanding. The numerical results demonstrate the practical superiority of the proposed algorithm.

**Weaknesses:**

1. line 65 typo "oerdering"
4. for better presentation, the balancing routine could be written as an algorithm instead of a definition
5. The authors need to better explain the balancing routine
6. The algorithm is complicated, the authors should provide with more motivation, remarks, and discussion about the design of algorithm.
7. Please provide a detailed analysis of the time and space complexity of solving the balancing problem, or to compare its complexity to existing methods.
8. typo at the end of line 19 of Algorithm 1
9. typo in Assumption 3.3
10. In the statement of Theorem 3.6 and 3.9, the definitions for $w$ and $\sigma$ are missing.
11. As a remark of Theorem 3.9, the authors might want to do discuss about the step size (depending on T). As T going large, the step size is diminishing. But why the converge rate is similar compare to Theorem 3.6 where a fixed step size is used? Please provide a detailed explanation of this apparent discrepancy, possibly including a comparison of the convergence behaviors under different step size regimes.
12. Please provide interpretations of the third and fourth terms in Theorem 3.9 in the context of the algorithm's behavior or performance. You could also suggest they discuss how these terms relate to key aspects of the method.
13. There is a gap between (2) and (13), so as Theorem 3.9 and its proof. I'm convinced with the proof and the theorem statement.
14. The authors need to discuss about the computation cost for the balance problem in the numerical section to make a fair comparison.

**Questions:**

See weakness

---

> ### Author Response · Authors · 2024-11-27
> **Responses to reviewer ntx7**
>
> We would like to thank you for your recognition of our work as well as your valuable suggestions.
> Below we address your concerns point by point with detailed explanation and revision:
>
> > **W1.** line 65 typo "oerdering"
>
> > **W6.** typo at the end of line 19 of Algorithm 1
>
> > **W7.** typo in Assumption 3.3
>
> Thank you for pointing out these typos.
> We have fixed them and also thoroughly checked the other parts of the whole submission.
>
> > **W2.** for better presentation, the balancing routine could be written as an algorithm instead of a definition
>
> Thank you for your suggestion.  In
> section 3.2 of
> this revised version,
> we added a new
> Algorithm 2
> for the balancing routine
> $\texttt{Balancing}(s, g_{m,k,t})$.
> It uses a greedy procedure that works well in practice.
> Specifically, we compare the vector norm of $\|s+ g_{m,k,t}\|$ and $\| s-g_{m,k,t}\|$,
> where $s+ g_{m,k,t}$ corresponds to putting the gradient vector $g_{m,k,t}$ at the beginning,
> and $s- g_{m,k,t}$ corresponds to putting the gradient vector $g_{m,k,t}$ at the end.
> Then since the online vector balancing problem in
> Definition 3.1
> tries to minimize the norm of vector sums,
> we choose the sample order that can lead to the smallest norm,
> as is indicated by the value of $\epsilon_{m,k,t}$:
> $\epsilon_{m,k,t}=1$ means putting the gradient vector $g_{m,k,t}$ at the beginning,
> while $\epsilon_{m,k,t}=-1$ means putting the gradient vector $g_{m,k,t}$ at the end.
>
> > **W3.** The authors need to better explain the balancing routine
>
> Thank you for your suggestion.
> We have added more explanation on the motivation of using the balancing routine in line 64-65 in the introduction,
> as is highlighted in blue,
> and more details can be found in highlighted parts in Section 3.2.
> Briefly speaking, solving the online vector balancing problem allows us to control the maximum norm of model update sums within one epoch,
> ($A$ in Assumption 3.7 and Proposition 3.8),
> which is related to the convergence rate as is shown in Theorem 3.9.
> Then by solving the online vector balancing problem,
> we try to make maximum norm of parameter updates in one epoch smaller,
> which then leads to faster convergence rate.
> Similar intuition can also be found in the paper of GraB (Grab: Finding provably better data permutations than random reshuffling. NeurIPS 2022).
> Please also refer to our responses to **W2** above for detailed procedure of the balancing routine.
>
> > **W4.** The algorithm is complicated, the authors should provide with more motivation, remarks, and discussion about the design of algorithm.
>
> Thank you for your suggestion.
> In Section 3.2 of this revised version,
> we added
> more explanations on the design of the proposed JoGBa.
> Specifically,
> JoGBa
> follows the same procedure as existing gradient-based multi-objective optimization algorithms.
> The difference starts at step 8 where the sample is selected by an order
> (which can be different for different objectives).
> Steps 11-16 then find the optimal order of samples
> based on the online vector balancing problem.
> Please also refer to our responses to **W2** for detailed procedure of the balancing routine and responses to **W3** for the motivation of gradient balancing.
>
> > **W5.** Please provide a detailed analysis of the time and space complexity of solving the balancing problem, or to compare its complexity to existing methods.
>
> > **W12.** The authors need to discuss about the computation cost for the balance problem in the numerical section to make a fair comparison.
>
> The newly added Algorithm 2 (that is used to solve the online vector balancing
> problem)
> only has a
> time
> complexity of $O(d)$
> (computing the sum, difference and the vector norm of two $d$-dimensional vectors),
> where $d$ is the dimensionality
> of the model parameter $w$.
> This is much cheaper than each
> model training iteration (computing model gradient and model update, indexed by $k$ in Algorithm 1),
> which can take $O(d^2)$ time
> (from
> "The Fine-Grained Complexity of Gradient Computation for Training Large Language
> Models", NeurIPS 2024).
>
> To further justify this,
> we have added a new Section 4.3 (highlighted in blue) to compare the time cost of each component on different multi-objective optimization problems.
> As is shown in the new Table 3, the per-iteration time cost of the balancing routine is almost negligible compared to the model update part.
>
> > **W8.** In the statement of Theorem 3.6 and 3.9, the definitions for $w$ and $\sigma$ are missing.
>
> In the previous version, their definitions are buried in the main text.
> In this revised version, we state them more explicitly in
> Theorem 3.6:
> (highlighted in blue).
> Specifically, $w^{(1)}_t$
> denotes the model parameter
> at training epoch $t$ and iteration 1,
> while $\sigma^2$ is defined as $\max_m \sigma^2_m$,
> where $\sigma^2_m$ is defined in Assumption 3.5 and is the gradient variance of the $m$-th objective.

---

> > ### Author Response · Authors · 2024-11-27
> > **Responses to reviewer ntx7 (cont.)**
> >
> > > **W9-1.** As a remark of Theorem 3.9, the authors might want to do discuss about the step size (depending on T). As T going large, the step size is diminishing. But why the converge rate is similar compare to Theorem 3.6 where a fixed step size is used? Please provide a detailed explanation of this apparent discrepancy,
> >
> > There is an error in step size $\alpha$ in Theorem 3.6 of the previous version. We have fixed it in this revised version, and as is highlighted in blue.
> > Theorem 3.6 should also use a diminishing step size
> > $\alpha=\sqrt{\frac{2 \Delta}{F_1 (F^2+\sigma^2)
> > T}}$, and is consistent with
> > the
> > stochastic optimization literature (e.g., "Stochastic First- and Zeroth-Order Methods for
> > Nonconvex Stochastic Programming". SIAM Journal on Optimization. 2013).
> >
> > > **W9-2.** possibly including a comparison of the convergence behaviors under different step size regimes.
> >
> > Thank you for your suggestion. We note that sample ordering methods allow us to use a larger step size $\alpha$
> > in Theorem 3.9 compared to random ordering in Theorem 3.6.
> > This in turn leads to better convergence rate for JoGBa ($O(T^{-2/3})$), compared to
> > that of random ordering ($O(T^{-1/2})$ in Theorem 3.6).
> > When using the same step size, the convergence rate of JoGBa reduces to the same $O(T^{-1/2})$ as the random ordering baseline.
> >
> > > **W10.** Please provide interpretations of the third and fourth terms in Theorem 3.9 in the context of the algorithm's behavior or performance. You could also suggest they discuss how these terms relate to key aspects of the method.
> >
> > We suppose the reviewer means the third and fourth terms on the right hand side of the bound.
> > These two terms are unique to methods with sample ordering
> > as we use a different step size $\alpha$,
> > and similar terms can also be found from the theoretical analysis in GraB (Grab: Finding provably better data permutations than random reshuffling. NeurIPS 2022).
> > Also, since both of them are $O(T^{-1})$ (note that $K$ becomes a constant for a given data set)
> > while the first term on the right hand side for Theorem 3.9 is $O(T^{-2/3})$,
> > only the first term dominates the convergence rate.
> > We have also highlighted the revised convergence results and discussions in blue in Section 3.3 to make it easier to interpret.
> >
> > > **W11.** There is a gap between (2) and (13), so as Theorem 3.9 and its proof. I'm convinced with the proof and the theorem statement.
> >
> > Thank you for pointing this out.
> > To derive the convergence rate
> > (2) in Theorem 3.6 from (13) in Appendix C.1 of this revised version,
> > we only need to set $\alpha=\sqrt{\frac{2 \Delta}{F_1 (F^2+\sigma^2) KT}}$ and the derivation follows by direct computation.
> > We have also revised the theorem statement and the proof in the Appendix to avoid any possible confusion.

---

> ### Author Response · Authors · 2024-12-03
>
> Dear reviewer ntx7,
>
> Thank you again for your valuable comments. As the discussion period is approaching its deadline, could you kindly take a moment to check our responses and let us know if we have adequately addressed your previous concerns? We would greatly appreciate any further feedback or comments you may have, and we are committed to addressing any outstanding issues that may have arisen.
>
> Best,
>
> Authors

---

### Official Review · Reviewer_wbXj · 2024-10-28

**Soundness:** 3
**Presentation:** 4
**Contribution:** 3
**Rating:** 8
**Confidence:** 3

**Summary:**

The paper proposes a novel sample order approach for multi-objective optimization problems. The method is inspired by the online vector balancing problem which tries to make the average of the centered gradient as close as to zero. Compared with different existing sample order approaches, the author shows that the proposed algorithm can achieve faster convergence. The experimental results on the NYUv2 and QM9 data sets also support the theory because the proposed algorithm outperforms different baseline methods.

**Strengths:**

Novel Approach to Multi-Objective Optimization: Inspired by the online vector balancing problem, The JoGBa method introduces a unique framework for ordering samples in multi-objective optimization by jointly balancing gradients across objectives . This is a new contribution to the multi-objective optimization field.

Theoretical Convergence Rate: The paper provides a thorough theoretical analysis. The convergence proofs are grounded in established assumptions, lending credibility to the claims.

Extensive Empirical Validation: JoGBa is validated on multiple datasets (e.g., NYUv2 and QM9) across diverse multi-objective optimization algorithms, such as MGDA, PCGrad, and Nash-MTL. This empirical diversity strengthens the generalizability of the findings.

**Weaknesses:**

Improved Convergence and Performance not clear: The authors claims that the method consistently outperforms existing data ordering strategies and dynamic weighting approaches. However, it seems the convergence rate for different algorithms are similar based on Theorem 3.6 and Theorem 3.7.

Presentation and Clarity: Certain sections, especially the theoretical parts and Algorithm 1, could be more reader-friendly. A more structured breakdown of steps and implications of theoretical results would enhance readability, making the technical details accessible to a broader audience.

Intuitions not sufficient: the online vector balancing problem is the core of balancing the gradient, but the authors fail to give enough intuition and explanation about how to implement the method across different objectives.

**Questions:**

1.	In line 189, the stale mean should be $\nu_{t+1}$. You write it as $m_{t+1}$. Please double check the notation.

2.	In Line 226, a necessary and sufficient condition for $\lambda$ not $x$.

3.	In Line 654, “beginning” instead of “begining”.

4.	In Definition 3.1, the authors mentioned online vector balancing which is the core of how to balance the gradient. If possible, please try to explain more clearly about the intuition (e.g., online vector balancing problem tries to make the average of the centered gradient as close as to “zero”). This will make the reader have a smooth reading experience.

5.	In Theorem 3.6 and Theorem 3.7, it seems the convergence rate of random sample ordering and the proposed algorithm are the same. Though in Theorem 3.6, the convergence rate is “=” while in Theorem 3.7, it is “<=”, one cannot conclude that the proposed algorithm converges faster than random sample ordering. The experimental results also show that the proposed algorithm converges just a little bit faster than other methods. Please try to explain clearly about the advantage of the proposed algorithm (maybe add some remarks/comments to give comparisons of different algorithms both theoretically and experimentally).

6.	If possible, let the reader know that the paper has a clear set up of the experiments in the appendix. This could help practitioners better understand the method’s sensitivity and optimize it for specific tasks.

---

> ### Author Response · Authors · 2024-11-27
> **Responses to reviewer wbXj**
>
> We would like to first thank you for your recognition on the contribution of our work for proposing a novel approach to multi-objective optimization as well as our theoretical and empirical results. For your concerns on the presentation of this paper (including several unclear parts in the previous version) and the intuition of our method, here we answer them point by point with detailed explanation:
>
> > **W1.** Improved Convergence and Performance not clear: The authors claims that the method consistently outperforms existing data ordering strategies and dynamic weighting approaches. However, it seems the convergence rate for different algorithms are similar based on Theorem 3.6 and Theorem 3.7.
>
> > **Q5-1.** In Theorem 3.6 and Theorem 3.7, it seems the convergence rate of random sample ordering and the proposed algorithm are the same. Though in Theorem 3.6, the convergence rate is “=” while in Theorem 3.7, it is “<=”, one cannot conclude that the proposed algorithm converges faster than random sample ordering.
>
> In this revised version,
> we use a
> larger step size $\alpha$
> in Theorem 3.9.
> The resultant asymptotic convergence rate of JoGBa is
> $O(T^{-2/3})$, which is
> faster than
> that of random ordering
> (with convergence rate $O(T^{-1/2})$ in Theorem 3.6).
> This is also verified by the empirical results in Figures 2-4.
>
> > **W2-1.** Presentation and Clarity: Certain sections, especially the theoretical parts and Algorithm 1, could be more reader-friendly.
>
> Thank you for your suggestion. In this revised version, we added clarifications
> (highlighted in blue)
> on the
> theoretical results
> and procedure
> of the proposed JoGBa.
> For the theoretical results, we clarified
> the definitions of several notations and added more detailed comparison on the
> convergence results of JoGBa (in Theorem 3.9) and random ordering (in Theorem 3.6).
> Please also refer to our responses to **W1** for a detailed comparison on the
> theoretical advantages of JoGBa over random ordering.
> For
> the
> JoGBa
> procedure
> (Algorithm 1),
> we added
> clarifications
> on the symbol superscripts and subscripts to help better understand the proposed method.
> We also added
> a new Algorithm 2,
> which shows how
> to solve
> the online vector balancing problem
> as a subroutine in Algorithm 1.
>
> > **W2-2.** A more structured breakdown of steps and implications of theoretical results would enhance readability, making the technical details accessible to a broader audience.
>
> Thank you for your suggestion. In
> Appendix C of
> this revised version, we added
> more detailed steps on the convergence analysis for both JoGBa and the random ordering baseline.
> Some implications from our theoretical results are discussed in Section
> 3.3,
> and we have highlighted them in blue for your easy reference.
>
> > **W3-1.** Intuitions not sufficient: the online vector balancing problem is the core of balancing the gradient, but the authors fail to give enough intuition
>
> > **Q4.** In Definition 3.1, the authors mentioned online vector balancing which is the core of how to balance the gradient. If possible, please try to explain more clearly about the intuition (e.g., online vector balancing problem tries to make the average of the centered gradient as close as to “zero”). This will make the reader have a smooth reading experience.
>
> As suggested, in this revised version we added more motivation
> on why online vector balancing is useful for accelerating convergence
> at the beginning of Section 3.2
> (highlighted in blue).
> The intuition
> is to control the maximum norm of parameter updates in one epoch ($A$ in Assumption 3.7 and Proposition 3.8),
> which is related to the convergence rate (Theorem 3.9).
> By solving the online vector balancing problem,
> we make the maximum norm of parameter updates in one epoch small,
> which then leads to faster convergence rate.
> Similar intuition can also be found in "Grab: Finding provably better data permutations than random reshuffling", NeurIPS 2022.
>
> On
> how to perform gradient balancing,
> we added
> its procedure in  the
> new Algorithm 2.
> Specifically, we use a simple greedy procedure to solve the online vector balancing problem in Definition 3.1.
> First, we compare the norms of $||s+ g_{m,k,t}||$ and $|| s-g_{m,k,t}||$,
> where $s+ g_{m,k,t}$ corresponds to putting the gradient vector $g_{m,k,t}$ at the beginning,
> and $s- g_{m,k,t}$ corresponds to putting the gradient vector $g_{m,k,t}$ at the end.
> Then since the online vector balancing problem in Definition 3.1 tries to minimize the norm of vector sums,
> we choose the sample order that leads to the smaller norm,
> as is indicated by the value of $\epsilon_{m,k,t}$:
> $\epsilon_{m,k,t}=1$ means putting the gradient vector $g_{m,k,t}$ at the beginning,
> while $\epsilon_{m,k,t}=-1$ means putting the gradient vector $g_{m,k,t}$ at the end.

---

> > ### Author Response · Authors · 2024-11-27
> > **Responses to reviewer wbXj (cont.)**
> >
> > > **W3-2.** and explanation about how to implement the method across different objectives.
> >
> > Regarding how we conduct gradient balancing across different objectives,
> > We have added more description (highlighted in blue) in Algorithm 1.
> > We first compute the gradient on the different objectives,
> > and apply the balancing routine (shown in the newly added Algorithm 2) with the same online sum vector $s$ on all these gradients to determine the sample order for different objectives.
> >
> > > **Q1.** In line 189, the stale mean should be $\nu_{t+1}$. You write it as $m_{t+1}$. Please double check the notation.
> >
> > > **Q2.** In Line 226, a necessary and sufficient condition for $\lambda$ not $x$.
> >
> > > **Q3.** In Line 654, “beginning” instead of “begining”.
> >
> > Thank you for pointing out the typos.
> > We have fixed them and also thoroughly checked the other parts of the whole submission.
> >
> > > **Q5-2.** The experimental results also show that the proposed algorithm converges just a little bit faster than other methods. Please try to explain clearly about the advantage of the proposed algorithm (maybe add some remarks/comments to give comparisons of different algorithms both theoretically and experimentally).
> >
> > Experimental results in
> > Figures 2(i)-(l) show
> > that JoGBa
> > has clearly
> > faster convergence
> > on NYUv2's
> > surface normal prediction task.
> > Similarly,
> > results in Tables 1 and 2 demonstrate that JoGBa
> > has better testing performance than
> > existing sample ordering methods.
> > On the theoretical side, Theorem 3.9 shows
> > that JoGBa
> > converges at a
> > rate of
> > $O(T^{-2/3})$,
> > which is
> > faster than
> > the
> > $O(T^{-1/2})$
> > rate
> > of random ordering
> > (in Theorem 3.6).
> >
> > > **Q6.** If possible, let the reader know that the paper has a clear set up of the experiments in the appendix. This could help practitioners better understand the method’s sensitivity and optimize it for specific tasks.
> >
> > Thank you for your suggestion.
> > Appendix A describes
> > the experimental setup, and
> > more details are now added in this revised version.

---

> > > ### Comment · Reviewer_wbXj · 2024-11-27
> > > **Reply to the Authors**
> > >
> > > 1. I see that the asymptotic convergence rate of the proposed algorithm is now better than the benchmark.
> > >
> > > 2. Thanks for adding more motivation on why online vector balancing is useful for accelerating convergence.
> > >
> > > 3. I also understand the experimental results. It would be great if the author could explain how the figures are related to $O(T^{-2/3})$ and $O(T^{-1/2})$ for different algorithms.
> > >
> > > Thanks the authors for the revision and the detailed explanation. All my concerns have been resolved.

---

> > > > ### Author Response · Authors · 2024-11-28
> > > > **Thanks and some further responses**
> > > >
> > > > We are glad to know that our responses resolved your previous concerns.
> > > > Regarding explaining how the figures are related to $O(T^{-2/3})$ and $O(T^{-1/2})$,
> > > > we have provided additional figures (Figure 5) in newly added Appendix D to better show how the weighted gradient norm decreases with the number of epochs.
> > > > By taking the logarithm of both the weighted gradient norm and the number of epochs $T$
> > > > (i.e., using a log-log plot),
> > > > we can see that empirical results on both random ordering and our method JoGBa matches our theoretical results in Section 3.3 well,
> > > > which also demonstrate that JoGBa has faster convergence rate compared to the random ordering baseline.

---

### Official Review · Reviewer_M4jx · 2024-11-08

**Soundness:** 3
**Presentation:** 3
**Contribution:** 4
**Rating:** 8
**Confidence:** 2

**Summary:**

The paper proposes a novel framework JoGBa for sample ordering in finite-sum multi-objective optimization problem, and demonstrate its empirical effectiveness on various MOO methods.

**Strengths:**

Data ordering for better optimization performance (that can especially outperform random shuffling) is important, and I think bring this concept to multi-objective optimization is an interesting and important contribution.

The paper theoretically analyze the convergence of MGDA for finite-sum MOO problem, with both random shuffling and JoGBa ordering.

Empirical results look promising.

**Weaknesses:**

For audience like me that is kind of familiar with MOO but not sample ordering, I think the presentation of algorithm 1 and figure 1 can be improved. Specifically,
(1) I cannot directly see from Figure 1, how the JoGBa approach (Fig 1 Right) is different from data ordering on each objective separately (Fig 1 Middle). In other words, Figure might not be illustrative enough.
(2) Algorithm 1 contains numerous superscripts and subscripts, making it challenging to interpret (which is acceptable if it prioritizes rigor). However, a clearer description might help—for example, one or two sentences explaining how JoGBa differs from ordering data for each objective separately. From the current description in lines 176 to 178, "The sample ordering is then determined based on the results of solving the balancing problem (routine Balancing) with the gradient on each objective," it’s still unclear (at least for me) how this approach differs from the one illustrated in the middle of Figure 1.

I believe the paper will be a clear accept for me, if the authors can clarify my concern or improve the presentation of the main methodology as mentioned above.

**Questions:**

See the question in previous section.

---

> ### Author Response · Authors · 2024-11-27
> **Responses to reviewer M4jx**
>
> We would like to first thank you for your recognition on the novelty and contribution of our work, including our theoretical and empirical results. For your concerns on the presentation of this paper, here we answer them point by point:
>
> > **W1-1.** how the JoGBa approach (Fig 1 Right) is different from data ordering on each objective separately (Fig 1 Middle). In other words, Figure might not be illustrative enough.
>
> > **W1-2.** a clearer description might help—for example, one or two sentences explaining how JoGBa differs from ordering data for each objective separately.
>
> > **W1-3.** From the current description in lines 176 to 178, "The sample ordering is then determined based on the results of solving the balancing problem (routine Balancing) with the gradient on each objective," it’s still unclear (at least for me) how this approach differs from the one illustrated in the middle of Figure 1.
>
> To better show the differences across the three sample ordering approaches in Figure 1,
> we further provide the detailed procedures of the two approaches shown in Figures 1(a) and 1(b),
> along with their differences with JoGBa (Figure 1(c)) in Appendix B.
> Specifically,
> the key difference
> between
> JoGBa
> and
> the approach in Figure 1(b)
> is that JoGBa uses the same sum vector $s$ to order gradients from the different objectives,
> while the approach in Figure 1(b) uses different vectors $s_m$'s for each objective $m=1, \dots, M$.
> This is a key novelty of this work. By performing online vector balancing jointly on different objectives,
> we can obtain a tighter bound on the maximum norm of parameter updates
> $|| w^{(k)}_t - w^{(1)}_t ||$
> (Proposition 3.8),
> while the
> upper bound for the
> approach in Figure 1(b)
> grows linearly with the number of objectives $M$
> (Proposition 3.10).
> The tighter bound then leads to faster
> convergence in Theorem 3.9
> (constant $A$ for
> JoGBa and
> $MA$ for the approach in Figure 1(b)).
> Empirical results
> are also provided
> in Figure 4 and Table 4,
> comparing the performance of JoGBa with ordering data for each objective separately.
> We can see that JoGBa outperforms all baselines that follow the approach in Figure 1(b),
> demonstrating the effectiveness of jointly balancing gradients from different objectives.
>
> > **W2.** Algorithm 1 contains numerous superscripts and subscripts, making it challenging to interpret (which is acceptable if it prioritizes rigor)
>
> Thank you for your suggestion.
> We have added more descriptions on these superscripts and subscripts in
> Algorithm 1, highlighted in blue.
> In summary, there are three different indices: $t$ is for the epoch number, $m$ is for the objective,
> and $k$ is for the iteration number in each epoch.

---

> > ### Comment · Reviewer_M4jx · 2024-11-28
> >
> > Thank you for the clarification and the additional information provided in the revision, I think my main concern is addressed.
> > I update my score to 8.

---

> > > ### Author Response · Authors · 2024-11-28
> > > **Thank you for your acknowledgement**
> > >
> > > We are glad to know that our responses addressed your previous concerns. Thank you again for your recognition of our work and increasing your rating score!

---

### Meta-Review · Area_Chair_LFhD · 2024-12-20

**Metareview:**

This paper considers the problem of developing a sample ordering approach for finite-sum multi objective optimization problems. The key insight behind the proposed approach (JoGBa) to balance gradients from multiple objectives is inspired by the online vector balancing problem which tries to make the average of the centered gradient as close as to zero. The paper develops theoretical analysis for JogBa to demonstrate that it can achieve faster convergence. Empirical results for JogBa are better than baseline methods.

All the reviewers' are positive about this paper and the author rebuttal addressed the questions/comments from reviewers.

Therefore, I recommend accepting the paper and strongly encourage the authors' to incorporate all the discussion in the camera copy to further improve the paper.

**Additional Comments On Reviewer Discussion:**

Most of the questions and comments were meant for clarifications and to constructively improve the paper.

Authors' have answered these questions and revised the paper.

Reviewer ntx7 did not respond to authors' rebuttal. I have gone through the author responses and they are satisfactory.

---

### Decision · Program_Chairs · 2025-01-22

Accept (Spotlight)